# Tight convex relaxations for sparse matrix factorization

**Emile Richard**
Electrical Engineering
Stanford University

**Guillaume Obozinski**
Université Paris-Est
Ecole des Ponts - ParisTech

**Jean-Philippe Vert**
MINES ParisTech
Institut Curie

## Abstract

Based on a new atomic norm, we propose a new convex formulation for sparse matrix factorization problems in which the number of non-zero elements of the factors is assumed fixed and known. The formulation counts sparse PCA with multiple factors, subspace clustering and low-rank sparse bilinear regression as potential applications. We compute slow rates and an upper bound on the statistical dimension [1] of the suggested norm for rank 1 matrices, showing that its statistical dimension is an order of magnitude smaller than the usual $\ell_1$-norm, trace norm and their combinations. Even though our convex formulation is in theory hard and does not lead to provably polynomial time algorithmic schemes, we propose an active set algorithm leveraging the structure of the convex problem to solve it and show promising numerical results.

## 1 Introduction

A range of machine learning problems such as link prediction in graphs containing community structure [16], phase retrieval [5], subspace clustering [18] or dictionary learning [12] amount to solve sparse matrix factorization problems, *i.e.*, to infer a low-rank matrix that can be factorized as the product of two sparse matrices with few columns (left factor) and few rows (right factor). Such a factorization allows more efficient storage, faster computation, more interpretable solutions and especially leads to more accurate estimates in many situations. In the case of interaction networks, for example, this is related to the assumption that the network is organized as a collection of highly connected communities which can overlap. More generally, considering sparse low-rank matrices combines two natural forms of sparsity, in the spectrum and in the support, which can be motivated by the need to explain systems behaviors by a superposition of latent processes which only involve a few parameters. Landmark applications of sparse matrix factorization are sparse principal components analysis (SPCA) [8, 21] or sparse canonical correlation analysis (SCCA)[19], which are widely used to analyze high-dimensional data such as genomic data.

In this paper, we propose new convex formulations for the estimation of sparse low-rank matrices. In particular, we assume that the matrix of interest should be factorized as the sum of rank one factors that are the product of column and row vectors with respectively $k$ and $q$ non zero-entries, where $k$ and $q$ are known. We first introduce below the $(k, q)$-rank of a matrix as the minimum number of left and right factors, having respectively $k$ and $q$ non-zeros, required to reconstruct a matrix. This index is a more involved complexity measure for matrices than the rank in that it conditions on the number of non-zero elements of the left and right factors of the matrix. Based on this index, we propose a new atomic norm for matrices [7] by considering its convex hull restricted to the unit ball of the operator norm, resulting in convex surrogates to low $(k, q)$-rank matrix estimation problem. We analyze the statistical dimension of the new norm and compare it to that of linear combinations of the $\ell_1$ and trace norms. In the vector case, our atomic norm actually reduces to $k$-support norm introduced by [2] and our analysis shows that its statistical power is not better than that of the $\ell_1$-

norm. By contrast, in the matrix case, the statistical dimension of our norm is at least one order of magnitude better than combinations of the $\ell_1$-norm and the trace norm.

However, while in the vector case the computation remains feasible in polynomial time, the norm we introduce for matrices can not be evaluated in polynomial time. We propose algorithmic schemes to approximately learn with the new norm. The same norm and meta-algorithms can be used as a regularizer in supervised problems such as multitask learning or quadratic regression and phase retrieval, highlighting the fact that our algorithmic contribution does not consist in providing more efficient solutions to the rank-1 SPCA problem, but to combine atoms found by the rank-1 solvers in a principled way.

## 2 Tight convex relaxations of sparse factorization constraints

In this section we propose a new matrix norm allowing to formulate various sparse matrix factorization problems as convex optimization problems. We start by defining the $(k, q)$-rank of a matrix in section 2.1, a useful generalization of the rank which also quantifies the sparseness of a matrix factorization. We then introduce in section 2.2 the $(k, q)$-trace norm, an atomic norm defined as the convex relaxations of the $(k, q)$-rank over the operator norm ball. We discuss further properties and potential applications of this norm used as a regularizer in section 2.3.

### 2.1 The $(k, q)$-rank of a matrix

The rank of a matrix $Z \in \mathbb{R}^{m_1 \times m_2}$ is the minimum number of rank-1 matrices needed to express $Z$ as a linear combination of the form $Z = \sum_{i=1}^{r} a_i b_i^\top$. The following definition generalizes this rank to incorporate conditions on the sparseness of the rank-1 elements:

**Definition 1 ($(k, q)$-sparse decomposition and $(k, q)$-rank)** *For a matrix $Z \in \mathbb{R}^{m_1 \times m_2}$, we call $(k, q)$-sparse decomposition of $Z$ any decomposition of the form $Z = \sum_{i=1}^{r} c_i a_i b_i^\top$ where $a_i$ (resp. $b_i$) are unit vectors with at most $k$ (resp. $q$) non-zero elements, and with minimal $r$, which we call the $(k, q)$-rank of $Z$.*

The $(k, q)$-rank and $(k, q)$-sparse decomposition of $Z$ can equivalently be defined as the optimal value and a solution of the optimization problem:

$$\min \|c\|_0 \quad \text{s.t.} \quad Z = \sum_{i=1}^{\infty} c_i a_i b_i^\top, \qquad (a_i, b_i, c_i) \in \mathcal{A}_k^{m_1} \times \mathcal{A}_q^{m_2} \times \mathbb{R}_+ \,, \tag{1}$$

where for any $1 \le j \le n$, $\mathcal{A}_j^n = \{a \in \mathbb{R}^n \mid \|a\|_0 \le j, \|a\|_2 = 1\}$. Since $\mathcal{A}_i^n \subset \mathcal{A}_j^n$ when $i \le j$, we have for any $k$ and $q$ $\text{rank}(Z) \le (k, q)\text{-rank}(Z) \le \|Z\|_0$. The $(k, q)$-rank is useful to formalize problems such as sparse matrix factorization, which can be defined as approximating the solution of a matrix valued problem by a matrix having low $(k, q)$-rank. For instance the standard rank-1 SPCA problem consists in finding the symmetric matrix with $(k, k)$-rank equal to 1 and providing the best approximation of the sample covariance matrix [21].

### 2.2 A convex relaxation for the $(k, q)$-rank

The $(k, q)$-rank is a discrete, nonconvex index, like the rank or the cardinality, leading to computational difficulties if one wants to learn matrices with small $(k, q)$-rank. We propose a convex relaxation of the $(k, q)$-rank aimed at mitigating these difficulties. For that purpose, we consider an atomic norm [7] that provides a convex relaxation of the $(k, q)$-trace norm, just like the $\ell_1$ norm and the trace norm are convex relaxations of the $\ell_0$ semi-norm and the rank, respectively. An atomic norm is a convex function defined based on a *small* set of elements called *atoms* which constitute a basis on which an object of interest can be sparsely decomposed. The function (a norm if the set is centrally symmetric) is defined as the gauge of the convex hull of atoms. In other terms, its unit ball or level-set of value 1 is formed by the convex envelope of atoms. In case of atoms of interest, namely rank-1 factors of given sparsities $k$ and $q$, we define

**Definition 2 ($(k, q)$-trace norm)** *Let $\mathcal{A}_{k,q}$ be a set of atoms $\mathcal{A}_{k,q} = \left\{ ab^\top \ : \ a \in \mathcal{A}_k^{m_1}, \ b \in \mathcal{A}_q^{m_2} \right\}$. For a matrix $Z \in \mathbb{R}^{m_1 \times m_2}$, the $(k, q)$-trace norm $\Omega_{k,q}(Z)$ is the atomic norm induced by $\mathcal{A}_{k,q}$, i.e.,*

$$\Omega_{k,q}(Z) = \inf \left\{ \sum_{A \in \mathcal{A}_{k,q}} c_A \ : \ Z = \sum_{A \in \mathcal{A}_{k,q}} c_A A, \ c_A \geq 0, \ \forall A \in \mathcal{A}_{k,q} \right\}. \tag{2}$$

In words, $\mathcal{A}_{k,q}$ is the set of matrices $A \in \mathbb{R}^{m_1 \times m_2}$ such that $(k,q)$-rank$(A) = 1$ and $\|A\|_{\mathrm{op}} = 1$. The next lemma provides an explicit formulation for the $(k,q)$-trace norm and its dual:

**Lemma 1** *For any* $Z, K \in \mathbb{R}^{m_1 \times m_2}$, *and denoting* $\mathcal{G}_k^m = \{I \subset [\![1,m]\!] \ : \ |I| = k\}$, *we have*

$$\Omega_{k,q}(Z) = \inf \left\{ \sum_{(I,J) \in \mathcal{G}_k^{m_1} \times \mathcal{G}_q^{m_2}} \|Z^{(I,J)}\|_* \ : \ Z = \sum_{(I,J)} Z^{(I,J)} \ , \ supp(Z^{(I,J)}) \subset I \times J \right\}, \tag{3}$$

*and* $\quad \Omega_{k,q}^*(K) = \max \left\{ \|K_{I,J}\|_{\mathrm{op}} \ : \ I \in \mathcal{G}_k^{m_1} \ , \ J \in \mathcal{G}_q^{m_2} \right\}.$

### 2.3 Learning matrices with sparse factors

In this section, we briefly discuss how the $(k,q)$-trace norm norm can be used to formulate various problems involving the estimation of sparse low-rank matrices. A way to learn a matrix $Z$ with low empirical risk $\mathcal{L}(Z)$ and with low $(k,q)$-rank is to use $\Omega_{k,q}$ as a regularizer and minimize an objective of the form

$$\min_{Z \in \mathbb{R}^{m_1 \times m_2}} \mathcal{L}(Z) + \lambda \Omega_{k,q}(Z). \tag{4}$$

A number of problems can be formulated as variants of (4).

**Bilinear regression.** In bilinear regression, given two inputs $x \in \mathbb{R}^{m_1}$ and $x' \in \mathbb{R}^{m_2}$ one observes as output a noisy version of $y = x^\top Z x'$. Assuming that $Z$ has low $(k,q)$-rank means that the noiseless response is a sum of a small number of terms, each involving only a small number of features from either of the input vectors. To estimate within such a model from observations $(x_i, x_i', y_i)_{i=1,\dots,n}$ one can consider the following formulation, in which $\ell$ is a convex loss :

$$\min_{Z \in \mathbb{R}^{m_1 \times m_2}} \sum_i \ell \left( x_i^\top Z x_i', y_i \right) + \lambda \Omega_{k,q}(Z). \tag{5}$$

**Subspace clustering.** In subspace clustering, one assumes that the data can be clustered in such a way that the points in each cluster belong to a low dimensional space. If we have a design matrix $X \in \mathbb{R}^{n \times p}$ with each row corresponding to an observation, then the previous assumption means that if $X^{(j)} \in \mathbb{R}^{n_j \times p}$ is a matrix formed by the rows of cluster $j$, there exist a low rank matrix $Z^{(j)} \in \mathbb{R}^{n_j \times n_j}$ such that $Z^{(j)} X^{(j)} = X^{(j)}$. This means that there exists a block-diagonal matrix $Z$ such that $ZX = X$ and with low-rank diagonal blocks. This idea, exploited recently by [18] implies that $Z$ is a sum of low rank sparse matrices; and this property still holds if the clustering is unknown. We therefore suggest that if all subspaces are of dimension $k$, $Z$ may be estimated via

$$\min_{Z \in \mathbb{R}^{n \times n}} \Omega_{k,k}(Z) \quad \text{s.t.} \quad ZX = X \ .$$

**Sparse PCA.** One possible formulation of sparse PCA with multiple factors is the problem of approximation of an empirical covariance matrix $\hat{\Sigma}_n$ by a low-rank matrix with sparse factors. This suggests to formulate sparse PCA as follows:

$$\min_Z \left\{ \|\hat{\Sigma}_n - Z\|_{\mathsf{F}} \ : \ (k,k)\text{-rank}(Z) \leq r \ \text{and} \ Z \succeq 0 \right\}, \tag{6}$$

where $q$ is the maximum number of non-zero coefficients allowed in each principal direction. By contrast to sequential approaches that estimate the principal components one-by-one [11], this formulation requires to find simultaneously a set of complementary factors. If we require the decomposition of $Z$ to be a sum of positive semi-definite $(k,k)$-sparse rank one factors (which is a stronger assumption than assuming that $Z$ is p.s.d.), the positivity constraint on $Z$ is no longer necessary and a natural convex relaxation for (6) using another atomic norm (in fact only a gauge here) is

$$\min_{Z \in \mathbb{R}^{m \times m}} \|\hat{\Sigma}_n - Z\|_{\mathsf{F}}^2 + \lambda \Omega_{k,\succeq}(Z), \tag{7}$$

where $\Omega_{k,\succeq}$ is the gauge of the set of atoms $\mathcal{A}_{k,\succeq} := \{aa^\top, a \in \mathcal{A}_k^m\}$.

# 3 Performance of the $(k, q)$-trace norm for denoising

In this section, we consider the problem of denoising a low-rank matrix $Z^\star \in \mathbb{R}^{m_1 \times m_2}$ with sparse factors corrupted by additive Gaussian noise, that is noisy observations $Y \in \mathbb{R}^{m_1 \times m_2}$ of the form $Y = Z^\star + \sigma G$, where $\sigma > 0$ and $G$ is a random matrix with i.i.d. $\mathcal{N}(0, 1)$ entries. For a convex penalty $\Omega : \mathbb{R}^{m_1 \times m_2} \to \mathbb{R}$, we consider, for any $\lambda > 0$, the estimator

$$\hat{Z}_\Omega^\lambda = \arg\min_Z \frac{1}{2} \|Z - Y\|_{\mathsf{F}}^2 + \lambda \Omega(Z) \,. \tag{8}$$

The following result is a straightforward generalization to any norm $\Omega$ of the so-called *slow rates* that are well know for the $\ell_1$ norms and other norms such as the trace-norm (see e.g. [10]).

**Lemma 2** *If* $\lambda \geq \sigma \Omega^*(G)$   *then*   $\left\|\hat{Z}_\Omega^\lambda - Z^\star\right\|_{\mathsf{F}}^2 \leq 4\lambda \Omega(Z^\star) \,.$

To derive an upper bound in estimation error from these inequalities, and to keep the argument as simple as possible we consider the oracle[1] estimate $\hat{Z}_\Omega^{\text{Oracle}}$ equal to $\hat{Z}_\Omega^\lambda$ where $\lambda = \sigma \Omega^*(G)$. From Lemma 2 we immediately get

$$\mathbb{E} \left\|\hat{Z}_\Omega^{\text{Oracle}} - Z^\star\right\|_{\mathsf{F}}^2 \leq 4\sigma \, \Omega(Z^\star) \, \mathbb{E} \, \Omega^*(G) \,. \tag{9}$$

This upper bound can be computed for $Z^\star = ab^\top \in \mathcal{A}_{k,q}$ for different norms. In particular, for $\Omega(Z^\star)$, we have $\|ab^\top\|_1 \leq \sqrt{kq}$ and $\Omega_{k,q}(ab^\top) = \|ab^\top\|_* = 1$ which lead to the corollary:

**Corollary 1** *When* $Z^\star = ab^\top \in \mathcal{A}_{k,q}$ *is an atom, the expected errors of the oracle estimators* $\hat{Z}_{\Omega_{k,q}}^{Oracle}$, $\hat{Z}_1^{Oracle}$ *and* $\hat{Z}_*^{Oracle}$ *using respectively the* $(k, q)$*-trace norm, the* $\ell_1$ *norm and the trace norm are upper bounded as follows:*

$$\begin{aligned} \mathbb{E} \|\hat{Z}_{\Omega_{k,q}}^{Oracle} - Z^\star\|_{\mathsf{F}}^2 &\leq 8\,\sigma \, \left( \sqrt{k \log \frac{m_1}{k} + 2k} + \sqrt{q \log \frac{m_2}{q} + 2q} \right) \,, \\ \mathbb{E} \|\hat{Z}_1^{Oracle} - Z^\star\|_{\mathsf{F}}^2 &\leq 2\sigma \|Z^\star\|_1 \sqrt{2 \log(m_1 m_2)} \leq 2\sigma \sqrt{2kq \log(m_1 m_2)} \,, \\ \mathbb{E} \|\hat{Z}_*^{Oracle} - Z^\star\|_{\mathsf{F}}^2 &\leq 2\sigma(\sqrt{m_1} + \sqrt{m_2}) \,. \end{aligned} \tag{10}$$

When the smallest entry in absolute value of $a$ or $b$ is close to 0, then the expected error is smaller for $\hat{Z}_1^{\text{Oracle}}$, reaching $\sigma \sqrt{2 \log(m_1 m_2)}$ on $e_1 e_1^\top$ while not changing for the two other norms. But under the assumption that the smallest nonzero entries in absolute value of $a$ and $b$ are lower bounded by $c/\sqrt{kq}$ with $c$ a constant, the upper bound on the rates obtained for the $(k, q)$-trace norm is at least an order of magnitude larger than for the other norms. We report the order of magnitude of these upper bounds in Table 1 for $m_1 = m_2 = m$ and $k = q$ and assuming that nonzeros coefficients are lower bounded in magnitude by $c/\sqrt{kq}$.

Obviously the comparison of upper bounds is not enough to conclude to the superiority of $(k, q)$-trace norm and, admittedly, the problem of denoising considered here is a special instance of linear regression in which the design matrix is the identity, and, since this is a case in which the design is trivially incoherent, it is possible to obtain fast rates for decomposable norms such as the $\ell_1$ or trace norm [13]; however, the slow rates obtained are the same if instead of $Y$ a linear transformation of $Z$ with incoherent design is observed, or when the signal to recover is only weakly sparse, which is not the case for the fast rates. Moreover, Lemma 2 applies to matrices of any rank and Corollary 1 generalizes to rank greater than 1. We present in the next section more sophisticated results, based on bounds on the so-called statistical dimension of different norms [1].

# 4 A bound on the statistical dimension of the $(k, q)$-trace norm

The squared *Gaussian width* [7, and ref. therein] and the *statistical dimension* introduced recently by Amelunxen et al. [1], provide quantified estimation guarantees. The two quantities are equal

up to an additive term smaller than 1 and we thus present results only in terms of the statistical dimension. The sample complexity of exact recovery and robust recovery are characterized by this quantity [7]. It is also equal to the signal to noise ratio necessary for denoising [6] and demixing [1] (see supplementary section 3). The statistical dimension is defined as follows: if $T_\Omega(A)$ is the tangent cone of a matrix norm $\Omega : \mathbb{R}^{m_1 \times m_2} \to \mathbb{R}_+$ at $A$, then, the statistical dimension of $T_\Omega(A)$ is

$$\mathfrak{S}(Z, \Omega) := \mathbb{E}\left[ \left\| \Pi_{T_\Omega(Z)}(G) \right\|_{\mathsf{F}}^2 \right],$$

where $G \in \mathbb{R}^{m_1 \times m_2}$ is a random matrix with i.i.d. standard normal entries and $\Pi_{T_\Omega(Z)}(G)$ is the orthogonal projection of $G$ onto the cone $T_\Omega(Z)$. In this section, we compute an upper bound on the statistical dimension of $\Omega_{k,q}$ at an atoms $A$ of $\mathcal{A}_{k,q}$, which we will denote by $\mathfrak{S}(A, \Omega_{k,q})$, and compare it to results known for linear combinations of the $\ell_1$ and the trace norm of the form $\Gamma_\mu$ with

$$\forall \mu \in [0,1], \ \forall Z \in \mathbb{R}^{m_1 \times m_2}, \quad \Gamma_\mu(Z) := \frac{\mu}{\sqrt{kq}}\|Z\|_1 + (1 - \mu)\|Z\|_\star , \tag{11}$$

which are norms that have been used in the literature to infer sparse low-rank matrices [17]. The ability to recover the support of a sparse vector typically depends on the size of its smallest non-zero coefficient. For the recovery of a sparse rank 1 matrix, this motivates the following definition

**Definition 3** *Let* $A = ab^\top \in \mathcal{A}_{k,q}$ *with* $I_0 = supp(a)$ *and* $J_0 = supp(b)$. *Denote* $a_{\min}^2 = \min\limits_{i \in I_0} a_i^2$ *and* $b_{\min}^2 = \min\limits_{j \in J_0} b_j^2$. *We define the strength* $\gamma(a, b) \in (0, 1]$ *as* $\gamma(a, b) := (k\, a_{\min}^2) \wedge (q\, b_{\min}^2)$.

The strength of an atom takes the maximal value of 1 when $|a_i| = 1/\sqrt{k}, i \in I$ and $|b_j| = 1/\sqrt{q}, j \in J$ where $I$ and $J$ are the supports of $a$ and $b$. On the contrary, its strength is close to 0 as soon as one of its nonzero entries is close to zero. We can now present our main result: a bound on the statistical dimension of $\Omega_{k,q}$ on $\mathcal{A}_{k,q}$.

**Proposition 1** *For* $A = ab^\top \in \mathcal{A}_{k,q}$ *with strength* $\gamma = \gamma(a, b)$, *there exist universal constants* $c_1, c_2$, *independent of* $m_1, m_2, k, q$ *such that*

$$\mathfrak{S}(A, \Omega_{k,q}) \ \leq \ \frac{c_1}{\gamma^2}(k + q) + \frac{c_2}{\gamma}(k + q)\log(m_1 \vee m_2) .$$

Our proof, presented in the appendix, follows the scheme proposed in [7] and used for the trace norm and $\ell_1$ norm. However, $\Omega_{k,q}$ is not decomposable and requires some work to obtain precise upper bounds on various quantities.

Note first that $\mathfrak{S}$ must be larger than the number of degrees of freedoms of elements of $\mathcal{A}_{k,q}$ which is $k + q - 1$. So the bound could not possibly be improved beyond logarithmic factors, besides the logarithmic dependence on the dimension of the overall space is expected. To appreciate the result, it should be compared with the statistical dimension for the $\ell_1$-norm which scales as the product of the size of the support with the logarithm of the dimension of the ambient space, that is as $kq \log(m_1 m_2)$. Using Landau notation, we report in Table 1 the upper and lower bounds known for the statistical dimension of other norms in the case where $k = q$ and $m_1 = m_2 = m$. The rates are known exactly up to constants for the $\ell_1$ and the trace norm (see e.g. [1]). Of particular relevance is the comparison with norms of the form $\Gamma_\mu$ which have been introduced with the aim of improving over both the $\ell_1$-norm and the trace norm and have been the object of a significant literature [17, 15, 9]. Using theorem 3.2 in [15], we prove in appendix 4 a lower bound on the statistical dimension of $\Gamma_\mu$ of order $kq \wedge (m_1 + m_2)$ which holds for all values of $\mu$, and which show that, up to logarithmic factors, $\Omega_{k,q}$ is an order of magnitude smaller in term of $k \wedge q$.

In the right column of Table 1 we also report results in the vector case, that is, when $m_2 = q = 1$. In fact, in that case, $\Omega_{k,1}$ is exactly the $k$-support norm proposed by [2]. But the statistical dimension of that norm and the $\ell_1$ norm is the same as it is known that the rate $k \log \frac{p}{k}$ cannot be improved ([4]). So, perhaps surprisingly, there improvement in the matrix case but not in the vector case.

## 5 Algorithm

In this section, we present a working set algorithm that attempts to solve problem (4). Injecting the variational form (3) of $\Omega_{k,q}$ in (4) and eliminating the variable $Z$ from the optimization problem

| Matrix norm | $\mathfrak{S}$ | $\Omega(Z^\star)\,\mathbb{E}\,\Omega^*(G)$ | Vector norm | $\mathfrak{S}$ |
|---|---|---|---|---|
| $(k,q)$-trace | $\mathcal{O}(k\log m)$ | $(k\log\frac{m}{k})^{1/2}$ | $k$-support | $\Theta(k\log\frac{p}{k})$ |
| $\ell_1$ | $\Theta(k^2\,\log\frac{m}{k^2})$ | $(k^2\log m)^{1/2}$ | $\ell_1$ | $\Theta(k\log\frac{p}{k})$ |
| trace-norm | $\Theta(m)$ | $m^{1/2}$ | $\ell_2$ | $p$ |
| $\ell_1$ + trace-n. | $\Omega(k^2\wedge m)$ | $\mathcal{O}\big(m^{1/2}\wedge(k^2\log m)^{1/2}\big)$ | elastic net | $\Theta(k\log\frac{p}{k})$ |

Table 1: Scaling of the statistical dimension $\mathfrak{S}$ and of the upper bound $\Omega(Z^\star)\,\mathbb{E}\Omega^*(G)$ in estimation error (slow-rates) of different matrix norms for elements of $\mathcal{A}_{k,q}$ with *strength* (see Definition 3) lower bounded by a constant (or equivalently with nonzero coefficient lower bounded by $c/\sqrt{kq}$ for $c$ a constant). Leftmost columns: scalings for matrices with $k=q$, $m=m_1=m_2$; rightmost columns: scalings for vectors with $m_1=p$ and $m_2=q=1$. We use the notations $\Omega$ and $\Theta$ with $f=\Omega(g)$ meaning $g=\mathcal{O}(f)$ and $f=\Theta(g)$ to mean that both $g=\mathcal{O}(f)$ and $f=\mathcal{O}(g)$.

using $Z=\sum_{(I,J)\in\mathcal{S}}Z^{(IJ)}$, one obtains that, when $\mathcal{S}=\mathcal{G}_k^{m_1}\times\mathcal{G}_q^{m_2}$, problem (4) is equivalent to

$$\min_{Z^{(IJ)}\in\mathbb{R}^{m_1m_2}}\mathcal{L}\Big(\sum_{(I,J)\in\mathcal{S}}Z^{(IJ)}\Big)+\lambda\sum_{(I,J)\in\mathcal{S}}\|Z^{(IJ)}\|_*,\quad\text{s.t.}\quad\mathrm{Supp}(Z^{(IJ)})\subset I\times J,\ (I,J)\in\mathcal{S}.\quad(\mathcal{P}_{\mathcal{S}})$$

At the optimum of (4) however, most of the variables $Z^{(IJ)}$ are equal to zero, and the solution is the same as the solution obtained from $(\mathcal{P}_{\mathcal{S}})$ in which $\mathcal{S}$ is reduced to the set of non-zero matrices $Z^{(IJ)}$ obtained at optimality, that are often called the *active* components. We thus propose to solve problem (4) using a so-called working set algorithm which solves a sequence of problems of the form $(\mathcal{P}_{\mathcal{S}})$ for a growing sequence of working sets $\mathcal{S}$, so as to keep a small number of non-zero matrices $Z^{(IJ)}$ throughout. Problem $(\mathcal{P}_{\mathcal{S}})$ is solved easily using approximate block coordinate descent on the $(Z^{(IJ)})_{(I,J)\in\mathcal{S}}$ [3, Chap. 4] , which consists in iterating proximal operators of the trace norm on blocks $I\times J$. The principle of the working set algorithm is to solve problem $(\mathcal{P}_{\mathcal{S}})$ for the current working set $\mathcal{S}$ and to check whether a new component should be added. It can be shown that a component with support $I\times J$ should be added if and only if $\|[\nabla\mathcal{L}(Z)]_{IJ}\|_{\mathrm{op}}>\lambda$ for the current value of $Z$. If such a component is found, the corresponding $(I,J)$ pair is added in $\mathcal{S}$ and problem $(\mathcal{P}_{\mathcal{S}})$ is solved again. Given that for any component in $\mathcal{S}$, we have $\|[\nabla\mathcal{L}(Z)]_{IJ}\|_{\mathrm{op}}\le\lambda$ at the optimum of $(\mathcal{P}_{\mathcal{S}})$, the algorithm terminates if $\Omega^*_{k,q}(\nabla\mathcal{L}(Z))\le\lambda$.

The main difficulty is that $\Omega^*_{k,q}(K)=\max\{a^\top Kb\mid a\in\mathcal{A}_k^{m_1},\ b\in\mathcal{A}_q^{m_2}\}$, which is NP-hard to compute, since it reduces in particular to rank 1 sparse PCA when $k=q$ and $K$ is p.s.d.. This implies that determining when the algorithm should stop and which new component to add is hard. However, a significant amount of research has been carried out on sparse PCA recently, and we thus propose to leverage some of the recently proposed relaxations and heuristics to solve this rank 1 sparse PCA problem (see [8, 20] and references therein). In particular, the Truncated Power iteration (TPI) algorithm of [20] can easily be modified to compute $\Omega^*_{k,q}$ which corresponds to a generalization of the rank 1 sparse PCA in which in general $a\ne b$ and $k\ne q$.

In our numerical experiments, we used a variant of Truncated Power Iteration with multiple restarts, keeping track of the highest found variance. It should be noted that under RIP conditions on the matrix, [20] shows that the solution returned by TPI is guaranteed to solve the rank 1 sparse PCA problem. Also, even if TPI finds a pair $(I,J)$ which is suboptimal, adding it in $\mathcal{S}$ does not hurt as the algorithm might determine subsequently that it is not necessary. However TPI might fail to find some of the components violating the optimality conditions and terminate the algorithm early.

The proposed algorithm cannot be guaranteed to solve (4) if $\Omega^*_{k,q}$ is not computed exactly, but it exploits as much as possible the structure of the convex optimization problem to find a candidate solution. A similar active set algorithm can be designed to solve problems regularized by $\Omega_{k,\succeq}$.

Formulations regularized by the trace norm require to compute its proximal operator, and thus to compute an SVD. However, even when $m_1,m_2$ are large, solving $\mathcal{P}_{\mathcal{S}}$ involves the computation of trace norms of matrices of size only $k\times q$ and so the SVDs that need to be computed are fairly small. This means that the computational bottleneck of the algorithm is clearly in finding candidate supports. It has been proved [20] that, under some conditions, the problem can be solved in linear time. Multiple restarts allow to find good candidate supports in practice.

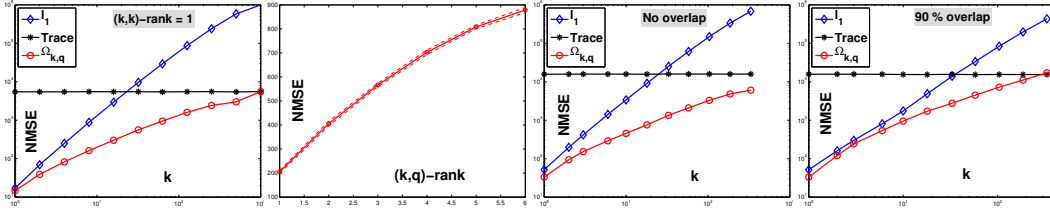

Figure 1: Estimates of the statistical dimensions of the $\ell_1$, trace and $\Omega_{k,q}$ norms at a matrix $Z \in \mathbb{R}^{1000 \times 1000}$ in different setting. From left to right: (a): $Z$ is an atom in $\widetilde{\mathcal{A}}_{k,k}$ for different values of $k$. (b) $Z$ is a sum of $r$ atoms in $\widetilde{\mathcal{A}}_{k,k}$ with non-overlapping support, with $k = 10$ and varying $r$, (c) $Z$ is a sum of 3 atoms in $\widetilde{\mathcal{A}}_{k,k}$ with non-overlapping support, for varying $k$. (d) $Z$ is a sum of 3 atoms in $\widetilde{\mathcal{A}}_{k,k}$ with overlapping support, for varying $k$.

## 6    Numerical experiments

In this section we report experimental results to assess the performance of sparse low-rank matrix estimation using different techniques. We start in Section 6.1 with simulations that confirm and illustrate the theoretical results on statistical dimension of $\Omega_{k,q}$ and assess how they generalize to matrices with $(k, q)$-rank larger than 1. In Section 6.2 we compare several techniques for sparse PCA on simulated data.

### 6.1    Empirical estimates of the statistical dimension.

In order to numerically estimate the statistical dimension $\mathfrak{S}(Z, \Omega)$ of a regularizer $\Omega$ at a matrix $Z$, we add to $Z$ a random Gaussian noise matrix and observe $Y = Z + \sigma G$ where $G$ has normal i.i.d. entries following $\mathcal{N}(0, 1)$. We then denoise $Y$ to form an estimate $\hat{Z}$ of $Z$. For small $\sigma$, the normalized mean-squared error (NMSE) defined as $\mathsf{NMSE}(\sigma) := \mathbb{E}\|\hat{Z} - Z\|_{\mathsf{F}}^2 / \sigma^2$ is a good estimate of the statistical dimension, since [14] show that $\mathfrak{S}(Z, \Omega) = \lim_{\sigma \to 0} \mathsf{NMSE}(\sigma)$. Numerically, we therefore estimate $\mathfrak{S}(Z, \Omega)$ with the empirical $\mathsf{NMSE}(\sigma)$ for $\sigma = 10^{-4}$, averaged over 20 replicates. We consider square matrices with $m_1 = m_2 = 1000$, and estimate the statistical dimension of $\Omega_{k,q}$, the $\ell_1$ and the trace norms at different matrices $Z$. The constrained denoiser has a simple closed-form for the $\ell_1$ and the trace norm. For $\Omega_{k,q}$, it can be obtained by a sequence of proximal projections with different parameters $\lambda$ until $\Omega_{k,q}(\hat{Z})$ has the correct value $\Omega_{k,q}(Z)$. Since the noise is small, we found that it was sufficient and faster to perform a $(k, q)$-SVD of $Y$ by computing a proximal of $\Omega_{k,q}$ with a small $\lambda$, and then apply the $\ell_1$ constrained denoiser to the set of $(k, q)$-sparse singular values.

We first estimate the statistical dimensions of the three norms at an atom $Z$ in $\widetilde{\mathcal{A}}_{k,q}$ for different values of $k = q$, where $\widetilde{\mathcal{A}}_{k,q} = \{ab^\top \in \mathcal{A}_{k,q} \mid \|ab^\top\|_\infty = 1/\sqrt{kq}\}$ is the set of elements of $\mathcal{A}_{k,q}$ with nonzero entries of constant magnitude . Figure 1.a shows the results, which confirm the theoretical bounds summarized in Table 1. The statistical dimension of the trace norm does not depend on $k$, while that of the $\ell_1$ norm increases almost quadratically with $k$ and that of $\Omega_{k,q}$ increases linearly with $k$. The linear versus quadratic dependence of the statistical dimension on $k$ are reflected by the slopes of the curves in the log-log plot in Figure 1.a. As expected, $\Omega_{k,q}$ interpolates between the $\ell_1$ norm (for $k = 1$) and the trace norm (for $k = m_1$), and outperforms both norms for intermediate values of $k$. This experiments therefore confirms that our upper bound (1) on $\mathfrak{S}(Z, \Omega_{k,q})$ captures the correct order in $k$, although the constants can certainly be much improved, and that our algorithm manages, in this simple setting, to correctly approximate the solution of the convex minimization problem.

Second, we estimate the statistical dimension of $\Omega_{k,q}$ on matrices with $(k, q)$-rank larger than 1, a setting for which we proved no theoretical result. Figure 1.b shows the numerical estimate of $\mathfrak{S}(Z, \Omega_{k,q})$ for matrices $Z$ which are sums of $r$ atoms in $\widetilde{\mathcal{A}}_{k,k}$ with non-overlapping support, for $k = 10$ and varying $r$. We observe that the increase in statistical dimension is roughly linear in the $(k, q)$-rank. For a fixed $(k, q)$-rank of 3, Figures 1.c and 1.d compare the estimated statistical dimensions of the three regularizers on matrices $Z$ which are sums of 3 atoms in $\widetilde{\mathcal{A}}_{k,k}$ with re-

| Sample covariance | Trace | $\ell_1$ | Trace + $\ell_1$ | Sequential | $\Omega_{k,\succeq}$ |
|---|---|---|---|---|---|
| $4.20 \pm 0.02$ | $0.98 \pm 0.01$ | $2.07 \pm 0.01$ | $0.96 \pm 0.01$ | $0.93 \pm 0.08$ | $\mathbf{0.59 \pm 0.03}$ |

Table 2: Relative error of covariance estimation with different methods.

spectively non-overlapping or overlapping supports. The shapes of the different curves are overall similar to the rank 1 case, although the performance of $\Omega_{k,q}$ degrades when the supports of atoms overlap. In both cases, $\Omega_{k,q}$ consistently outperforms the two other norms. Overall these experiments suggest that the statistical dimension of $\Omega_{k,q}$ at a linear combination of $r$ atoms increases as $Cr \left( k \log m_1 + q \log m_2 \right)$ where the coefficient $C$ increases with the overlap among the supports of the atoms.

## 6.2 Comparison of algorithms for sparse PCA

In this section we compare the performance of different algorithms in estimating a sparsely factored covariance matrix that we denote $\Sigma^\star$. The observed sample consists of $n$ i.i.d. random vectors generated according to $\mathcal{N}(0, \Sigma^\star + \sigma^2 \mathrm{Id}_p)$, where $(k,k)$-rank$(\Sigma^\star) = 3$. The matrix $\Sigma^\star$ is formed by adding 3 blocks of rank 1, $\Sigma^\star = a_1 a_1^\top + a_2 a_2^\top + a_3 a_3^\top$, having all the same sparsity $\|a_i\|_0 = k = 10$, $3 \times 3$ overlaps and nonzero entries equal to $1/\sqrt{k}$. The noise level $\sigma = 0.8$ is set in order to make the signal to noise ratio below the level $\sigma = 1$ where a spectral gap appears and makes the spectral baseline (penalizing the trace of the PSD matrix) work. In our experiments the number of variables is $p = 200$ and $n = 80$ points are observed. To estimate the true covariance matrix from the noisy observation, first the sample covariance matrix is formed as $\hat{\Sigma}_n = \frac{1}{n} \sum_{i=1}^{n} x_i x_i^\top$, and given as input to various algorithms which provide a new estimate $\hat{\Sigma}$. The methods we compared are the following:

• **Sample covariance.** Output $\hat{\Sigma}_n$ as the estimate of the covariance.

• $\ell_1$ **penalty.** Soft-threshold $\hat{\Sigma}_n$ elementwise.

• **Trace penalty on the PSD cone.** $\min_{Z \succeq 0} \frac{1}{2} \|Z - \hat{\Sigma}_n\|_{\mathsf{F}}^2 + \lambda \operatorname{Tr} Z$.

• **Trace + $\ell_1$ penalty.** $\min_{Z \succeq 0} \frac{1}{2} \|Z - \hat{\Sigma}_n\|_{\mathsf{F}}^2 + \lambda \Gamma_\mu(Z)$.

• $\Omega_{k,\succeq}$ **penalty.** $\min_{Z \in \mathbb{R}^{p \times p}} \frac{1}{2} \|Z - \hat{\Sigma}_n\|_{\mathsf{F}}^2 + \lambda \Omega_{k,\succeq}(Z)$, with $\Omega_{k,\succeq}$ the gauge associated with $\mathcal{A}_{k,\succeq}$ introduced in Section 2.3.

• **Sequential sparse PCA.** This is the standard way of estimating multiple sparse principal components which consists of solving the problem for a single component at each step $t = 1 \ldots r$, and deflate to switch to the next $(t+1)$st component. The deflation step used in this algorithm is the orthogonal projection $Z_{t+1} = \left( \mathrm{Id}_p - u_t u_t^\top \right) Z_t \left( \mathrm{Id}_p - u_t u_t^\top \right)$. The tuning parameters for this approach are the sparsity level $k$ and the number of principal components $r$. The hyperparameters were chosen by leaving one portion of the train data off (validation) and selecting the parameter which allows to build an estimator approximating the best the validation set's empirical covariance. We assumed the true value of $k$ known in advance for all algorithms.

We report the relative errors $\|\hat{\Sigma} - \Sigma^\star\|_{\mathsf{F}} / \|\Sigma^\star\|_{\mathsf{F}}$ over 10 runs of our experiments in Table 2. The results indicate that sparse PCA methods, whether based on $\Omega_{k,\succeq}$ or the sequential method with deflation steps, outperform spectral and $\ell_1$ baselines, and that penalizing $\Omega_{k,\succeq}$ is superior to the sequential approach. This was to be expected since our algorithm minimizes a loss function close to the error measure used, whereas the sequential scheme does not optimize a well-defined objective.

## 7 Conclusion

We formulated the problem of matrix factorization with sparse factors of known sparsity as the minimization of an index, the $(k,q)$-rank which tight convex relaxation is the $(k,q)$-trace norm regularizer. This penalty is proved to have near optimal statistical performance. Despite theoretical computational hardness in the worst-case scenario, exploiting the convex geometry of the problem allowed us to build an efficient algorithm to minimize it. Future work will consist of relaxing the constraint on the blocks size, and exploring applications such as finding small comminuties in large random graph background.

**Acknowlegments** This project was partially funded by Agence Nationale de la Recherche grant ANR-13-MONU-005-10 (CHORUS project) and by ERC grant SMAC-ERC-280032.

## Footnotes

[1]We call it oracle estimate because the choice of $\lambda$ depends on the unknown noise level. Virtually identical bounds (up to constants) holding with large probability could be derived for the non-oracle estimator by controlling the deviations of $\Omega^*(G)$ from its expectation.

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
