[Supplementary Material]

# Tight convex relaxations for sparse matrix factorization (Appendix)

**Emile Richard**
Electrical Engineering
Stanford University

**Guillaume Obozinski**
Université Paris-Est
Ecole des Ponts - ParisTech

**Jean-Philippe Vert**
MINES ParisTech
Institut Curie

## 1    Characterizations of $\Omega_{k,q}$ and its dual.

**Lemma 1.** *The $(k,q)$-SVD is not necessarily unique, and the factors are not necessarily orthogonal.*

*Proof of Lemma 1.* Consider the case of the $(2,2)$-SVD for the matrix $Z = \mathbf{1}\mathbf{1}^\top \in \mathbb{R}^3$. It is impossible to write $Z$ as the sum of two $(2,2)$-sparse matrices, because it would then have at most $8$ non-zero coefficients. But we have the decomposition.

$$\begin{pmatrix} 1 & 1 & 1 \\ 1 & 1 & 1 \\ 1 & 1 & 1 \end{pmatrix} = \begin{pmatrix} 2 & 1 & 0 \\ 1 & \frac{1}{2} & 0 \\ 0 & 0 & 0 \end{pmatrix} + \begin{pmatrix} 0 & 0 & 1 \\ 0 & \frac{1}{2} & 1 \\ 0 & 1 & 2 \end{pmatrix} - \begin{pmatrix} 1 & 0 & -1 \\ 0 & 0 & 0 \\ -1 & 0 & 1 \end{pmatrix},$$

which shows that the $(2,2)$-rank of $Z$ is 3. We see that the three rank-1 matrices in this decomposition are not orthogonal. In addition, given that $Z$ is invariant by any of the 6 permutations of the rows and any of the 6 permutations of the columns, $Z$ admits at least 36 different $(2,2)$-SVDs. $\square$

**Lemma 2.** *For any $Z, K \in \mathbb{R}^{m_1 \times m_2}$, and denoting $\mathcal{G}_k^m = \{I \subset [\![1,m]\!] \ : \ |I| = k\}$, we have*

$$\Omega_{k,q}(Z) = \inf \left\{ \sum_{(I,J) \in \mathcal{G}_k^{m_1} \times \mathcal{G}_q^{m_2}} \|Z^{(I,J)}\|_* \ : \ Z = \sum_{(I,J)} Z^{(I,J)} \ , \ supp(Z^{(I,J)}) \subset I \times J \right\}, \quad (1)$$

*and*

$$\Omega_{k,q}^*(K) = \max \left\{ \|K_{I,J}\|_{\mathrm{op}} \ : \ I \in \mathcal{G}_k^{m_1} \ , \ J \in \mathcal{G}_q^{m_2} \right\}. \quad (2)$$

*Proof of Lemma 2.* We first show (2) from the definition of the dual norm $\Omega_{k,q}^*$:

$$\begin{aligned} \Omega_{k,q}^*(K) &= \max_Z \left\{ \langle K, Z \rangle \ : \ \Omega_{k,q}(Z) \leq 1 \right\} \\ &= \max_{a,b} \left\{ \langle K, ab^\top \rangle \ : \ ab^\top \in \mathcal{A}_{k,q} \right\} \\ &= \max_{a,b} \left\{ a^\top K b \ : \ \|a\|_0 \leq k \ , \ \|b\|_0 \leq q \ , \ \|a\|_2 = \|b\|_2 = 1 \right\} \\ &= \max_{I,J} \left\{ \|K_{I,J}\|_{\mathrm{op}} \ : \ I \in \mathcal{G}_k^{m_1} \ , \ J \in \mathcal{G}_q^{m_2} \right\}, \end{aligned}$$

where the second equality follows from the fact that the maximization of a linear form over a bounded convex set is attained at one of the extreme points of the set.

Given this closed-form expression of the dual norm, we prove the variational formulation (1) for the primal norm $\Omega_{k,q}$. Consider the function $\check{\Omega}_{k,q}$ defined by

$$\check{\Omega}_{k,q}(Z) = \inf \left\{ \sum_{(I,J) \in \mathcal{G}_k^{m_1} \times \mathcal{G}_q^{m_2}} \|Z^{(I,J)}\|_* \ : \ Z = \sum_{(I,J)} Z^{(I,J)} \ , \ supp(Z^{(I,J)}) \subset I \times J \right\}.$$

Since $\check{\Omega}_{k,q}(Z)$ is defined as the infimum of a jointly convex function of $Z$ and $(Z^{(I,J)})_{I \in \mathcal{G}_k^{m_1}, J \in \mathcal{G}_q^{m_2}}$ obtained by minimizing w.r.t. to the latter variables, it is a an elementary fact from convex analysis

that $\check{\Omega}_{k,q}$ is a convex function of $Z$. It is also symmetric and positively homogeneous, which together with convexity prove that $\check{\Omega}_{k,q}$ defines a norm. We can compute its dual norm as

$$
\begin{aligned}
\check{\Omega}_{k,q}^*(K) &= \max_{Z} \left\{ \langle K, Z \rangle \ : \ \check{\Omega}_{k,q}(Z) \leq 1 \right\} \\[2mm]
&= \max_{(Z^{(IJ)})_{(I,J)}} \left\{ \langle K, \sum_{(I,J)} Z^{(IJ)} \rangle \ : \ \sum_{(I,J)} \|Z^{(IJ)}\|_* \leq 1 \quad , \quad \mathrm{supp}(Z^{(IJ)}) \subset I \times J \right\} \\[2mm]
&= \max_{(Z^{(IJ)})_{(I,J)}, (\eta^{(IJ)})_{(I,J)}} \left\{ \sum_{(I,J)} \eta^{(I,J)} \langle K_{I,J}, Z^{(IJ)} \rangle \ : \ \|Z^{(IJ)}\|_* \leq \eta^{(IJ)}, \quad \sum_{(I,J)} \eta^{(IJ)} \leq 1 \right\} \\[2mm]
&= \max_{(\eta^{(IJ)})_{(I,J)}} \left\{ \sum_{(I,J)} \eta^{(IJ)} \|K_{I,J}\|_{\mathrm{op}} \ : \ \sum_{(I,J)} \eta^{(IJ)} \leq 1 \right\} \\[2mm]
&= \max_{(I,J)} \|K_{I,J}\|_{\mathrm{op}} \\[2mm]
&= \Omega_{k,q}^*(K)
\end{aligned}
$$

This proves that $\Omega_{k,q}(K) = \check{\Omega}_{k,q}(K)$ since a norm is uniquely characterized by its dual norm. $\qquad\square$

## 2 Slow-rate bounds

### 2.1 Proof of Lemma 2 of the main manuscrit

We prove first a more general result. Let $\Omega : \mathbb{R}^{m_1 \times m_2} \to \mathbb{R}$ be any matrix norm, and $\mathcal{X} : \mathbb{R}^{m_1 \times m_2} \to \mathbb{R}^n$ be a linear map. We denote by $X_i$ $(i = 1, \ldots, n)$ the $i$-th design matrix defined by $\mathcal{X}(Z)_i = \langle Z, X_i \rangle$. For a given matrix $Z^\star \in \mathbb{R}^{m_1 \times m_2}$, assume we observe:

$$
Y = \mathcal{X}(Z^\star) + \epsilon, \tag{3}
$$

where $\epsilon \in \mathbb{R}^n$ is a centered random noise vector. We consider the following estimator of $Z^\star$:

$$
\hat{Z}_\Omega \in \arg\min_Z \frac{1}{2n} \|Y - \mathcal{X}(Z)\|_2^2 + \lambda \Omega(Z), \tag{4}
$$

for some value of the parameter $\lambda > 0$. The following result generalizes standard results known for the $\ell_1$ and trace norms (e.g., Theorem 1 in [9]) to any norm $\Omega$.

**Theorem 1.** *If $\lambda \geq \frac{1}{n} \Omega^*(\sum_{i=1}^n \epsilon_i X_i)$ then*

$$
\frac{1}{2n} \|\mathcal{X}(\hat{Z}_\Omega - Z^\star)\|_2^2 \leq \inf_Z \left\{ \frac{1}{2n} \|\mathcal{X}(Z - Z^\star)\|_2^2 + 2\lambda \Omega(Z) \right\}. \tag{5}
$$

*Proof of Theorem 1.* By definition of $\hat{Z}_\Omega$ (4), we have for all $Z$:

$$
\frac{1}{2n} \|Y - \mathcal{X}(\hat{Z}_\Omega)\|_2^2 \leq \frac{1}{2n} \|Y - \mathcal{X}(Z)\|_2^2 + \lambda \left( \Omega(Z) - \Omega(\hat{Z}_\Omega) \right),
$$

which after developing the squared norm and replacing $Y$ by (3) gives

$$
\frac{1}{2n} \|\mathcal{X}(\hat{Z}_\Omega)\|_2^2 - \frac{1}{n} \langle \mathcal{X}(Z^\star) + \epsilon, \mathcal{X}(\hat{Z}_\Omega) \rangle \leq \frac{1}{2n} \|\mathcal{X}(Z)\|_2^2 - \frac{1}{n} \langle \mathcal{X}(Z^\star) + \epsilon, \mathcal{X}(Z) \rangle + \lambda \left( \Omega(Z) - \Omega(\hat{Z}_\Omega) \right),
$$

and therefore

$$
\frac{1}{2n} \|\mathcal{X}(\hat{Z}_\Omega - Z^\star)\|_2^2 \leq \frac{1}{2n} \|\mathcal{X}(Z - Z^\star)\|_2^2 + \frac{1}{n} \langle \epsilon, \mathcal{X}(\hat{Z}_\Omega - Z) \rangle + \lambda \left( \Omega(Z) - \Omega(\hat{Z}_\Omega) \right). \tag{6}
$$

Now, using the fact (true for any norm) that $\Omega(A)\Omega^\star(B) \geq \langle A, B \rangle$ for any vectors $A, B \in \mathbb{R}^n$, and taking $\lambda \geq \frac{1}{n}\Omega^*(\sum_{i=1}^n \epsilon_i X_i)$, we can upper bound the second term of the right-hand side of (6) by:

$$
\begin{aligned}
\frac{1}{n}\langle \epsilon, \mathcal{X}(\hat{Z}_\Omega - Z)\rangle &= \frac{1}{n}\sum_{i=1}^n \epsilon_i \mathcal{X}(\hat{Z}_\Omega - Z)_i \\
&= \frac{1}{n}\sum_{i=1}^n \epsilon_i \langle X_i, \hat{Z}_\Omega - Z\rangle \\
&= \frac{1}{n}\langle \sum_{i=1}^n \epsilon_i X_i, \hat{Z}_\Omega - Z\rangle \\
&\leq \frac{1}{n}\Omega^\star\left(\sum_{i=1}^n \epsilon_i X_i\right)\Omega\left(\hat{Z}_\Omega - Z\right) \\
&\leq \lambda\Omega\left(\hat{Z}_\Omega - Z\right)
\end{aligned}
$$

Plugging this bound back in (6) finally gives

$$
\begin{aligned}
\frac{1}{2n}\|\mathcal{X}(\hat{Z}_\Omega - Z^\star)\|_2^2 &\leq \frac{1}{2n}\|\mathcal{X}(Z - Z^\star)\|_2^2 + \lambda\Omega(\hat{Z}_\Omega - Z) + \lambda\left(\Omega(Z) - \Omega(\hat{Z}_\Omega)\right) \\
&\leq \frac{1}{2n}\|\mathcal{X}(Z - Z^\star)\|_2^2 + 2\lambda\Omega(Z),
\end{aligned}
$$

the last inequality being due to the triangle inequality. $\square$

**Lemma 3.** *If* $\lambda \geq \sigma\Omega^*(G)$ *then* $\left\|\hat{Z}_\Omega^\lambda - Z^\star\right\|_F^2 \leq 4\lambda\Omega(Z^\star)$.

*Proof of Lemma 3.* This is a simple consequence of Theorem 1 by taking for $\mathcal{X}$ the identity map, upper bounding the right-hand side of (5) by the value $2\lambda\Omega(Z^\star)$ it takes for $Z = Z^\star$, and replacing $\lambda$ by $\lambda/n$. $\square$

## 2.2  Proof of Corollary 1 of the main manuscript

Corollary 1 of the main manuscript is a direct application of Equation 10 therein, once we have the following upper bounds on the dual norms of a random noise matrix $G$:

**Proposition 1.** *Let* $G \in \mathbb{R}^{m_1 \times m_2}$ *be a random matrix with entries i.i.d. from* $\mathcal{N}(0,1)$. *The expected dual norm of* $G$ *for the* $(k,q)$-*trace norm, the* $\ell_1$ *norm and the trace norm is respectively bounded by:*

$$
\begin{aligned}
\mathbb{E}\,\Omega_{k,q}^*(G) &\leq 4\left(\sqrt{k\log\frac{m_1}{k} + 2k} + \sqrt{q\log\frac{m_2}{q} + 2q}\right), \\
\mathbb{E}\,\|G\|_1^* &\leq \sqrt{2\log(m_1 m_2)}, \\
\mathbb{E}\,\|G\|_\star^* &\leq \sqrt{m_1} + \sqrt{m_2}.
\end{aligned}
\tag{7}
$$

To prove Proposition 1, let us start by the following

**Lemma 4.** *Let* $G$ *be a* $m_1 \times m_2$ *random matrix with i.i.d. normally distributed entries. Then*

$$
\mathbb{E}\max_{I\in\mathcal{G}_k, J\in\mathcal{G}_q}\|G_{I,J}\|_{\mathrm{op}}^2 \leq 16\left[\left(k\log\frac{m_1}{k} + q\log\frac{m_2}{q}\right) + 2(k+q)\right].
$$

*Proof of Lemma 4.* For a random matrix $H \in \mathbb{R}^{k\times q}$ with i.i.d. standard normal entries, we have the following concentration inequality [5]: for $s \geq 0$,

$$
\mathbb{P}[\|H\|_{\mathrm{op}} > \sqrt{k} + \sqrt{q} + s] \leq \exp(-s^2/2).
\tag{8}
$$

Denoting $R = 2\left(\sqrt{k} + \sqrt{q}\right)$, and $f(x) = e^{tx^2}$, we have the sequence of inequalities

$$\mathbb{E}\exp(t\|H\|_{\mathrm{op}}^2) = \mathbb{E}f(\|H\|_{\mathrm{op}})$$

$$= \int_1^\infty \mathbb{P}[f(\|H\|_{\mathrm{op}}) > h] \ dh$$

$$\leq \int_1^{1+f(R)} 1 \ dh + \int_{1+f(R)}^\infty \mathbb{P}[f(\|H\|_{\mathrm{op}}) > h]dh$$

$$= f(R) + \int_0^\infty \mathbb{P}[\|H\|_{\mathrm{op}} > f^{-1}(f(R) + 1 + \zeta)]d\zeta$$

$$\leq f(R) + \int_0^\infty \mathbb{P}[\|H\|_{\mathrm{op}} > \frac{1}{2}R + \frac{1}{2}f^{-1}(1+\zeta)]d\zeta \tag{9}$$

$$\leq f(R) + \int_0^\infty 8ts \ \exp\left(-s^2/2 + 4ts^2\right) ds \tag{10}$$

$$\leq f(R) + 4\frac{t}{\frac{1}{2} - 4t} \tag{11}$$

$$\leq \exp(8t(k+q)) + \frac{8t}{1 - 8t},$$

where the change of variable used in (10) is $1 + \zeta = f(2s) = e^{4ts^2}$, (11) is true for any $t < \frac{1}{8}$, and (9) follows from the property of the inverse $f^{-1}(z) = \sqrt{\frac{\log(z)}{t}}$ that it is strictly increasing on $[1; \infty)$ and sandwiched via

$$\frac{1}{2}\left\{f^{-1}(z) + f^{-1}(z')\right\} \leq f^{-1}(z + z') \leq f^{-1}(z) + f^{-1}(z'). \tag{12}$$

Take now $t = \frac{1}{8} - \frac{1}{8(k+q)}$. Since $k + q \geq 2$, we have $1/16 \leq t < 1/8$. Therefore,

$$\mathbb{E}\max_{I,J}\|G_{I,J}\|_{\mathrm{op}}^2 = \frac{1}{t}\log\left\{\exp t\,\mathbb{E}\max_{I,J}\|G_{I,J}\|_{\mathrm{op}}^2\right\}$$

$$\leq \frac{1}{t}\log\left\{\mathbb{E}\exp(t\max_{I,J}\|G_{I,J}\|_{\mathrm{op}}^2)\right\}$$

$$\leq \frac{1}{t}\log\left\{\sum_{I,J}\mathbb{E}\exp(t\|G_{I,J}\|_{\mathrm{op}}^2)\right\}$$

$$\leq \frac{1}{t}\log\left\{\binom{m_1}{k}\binom{m_2}{q}\mathbb{E}\exp(t\|H\|_{\mathrm{op}}^2)\right\}$$

$$\leq \frac{1}{t}\log\left\{\left(\frac{e\,m_1}{k}\right)^k\left(\frac{e\,m_2}{q}\right)^q\left(e^{8t(k+q)} + \frac{8t}{1-8t}\right)\right\}$$

$$= \frac{1}{t}\left[\left(k\log\frac{m_1}{k} + q\log\frac{m_2}{q}\right) + k + q + 8t(k+q) + \log\left(1 + \frac{8t}{1-8t}e^{-8t(k+q)}\right)\right]$$

$$\leq 16\left[\left(k\log\frac{m_1}{k} + q\log\frac{m_2}{q}\right) + k + q\right] + 8(k+q) + \frac{8}{1-8t}e^{-8t(k+q)}$$

$$\leq 16\left[\left(k\log\frac{m_1}{k} + q\log\frac{m_2}{q}\right) + 2(k+q)\right],$$

where in the last inequality we simply used $8/(1 - 8t) = 8(k + q)$ and $\exp(-8t(k + q)) \leq 1$. $\quad\square$

*Proof of Proposition 1.* The bounds on the dual norm of the $\ell_1$ and trace norms are standard, so we just focus on the bound for $\Omega_{k,q}$. Using Jensen's inequality, we easily get from Lemma 4:

$$
\begin{aligned}
\mathbb{E}\,\Omega_{k,q}^*(G) = \mathbb{E} \max_{I \in \mathcal{G}_k, J \in \mathcal{G}_q} \|G_{I,J}\|_{\mathrm{op}} \\
\leq \left( \mathbb{E} \max_{I \in \mathcal{G}_k, J \in \mathcal{G}_q} \|G_{I,J}\|_{\mathrm{op}}^2 \right)^{\frac{1}{2}} \\
\leq 4 \left[ \left( k \log \frac{m_1}{k} + q \log \frac{m_2}{q} \right) + 2(k+q) \right]^{\frac{1}{2}} \\
\leq 4 \left( \sqrt{k \log \frac{m_1}{k} + 2k} + \sqrt{q \log \frac{m_2}{q} + 2q} \right).
\end{aligned}
$$

$\square$

# 3   Results on statistical dimensions

Powerful results from asymptotic geometry have recently been used by [4, 12, 1, 6] to quantify the statistical power of a convex nonsmooth regularizer used as a constraint or penalty. These results rely essentially on the fact that if the cone of descent directions of the regularizer at a point of interest $Z$ is thiner, then the regularizer is more efficient at solving problems of denoising, demixing and estimation of $Z$. The gain in efficiency can be quantified by appropriate measures of width of the tangent cone such as the Gaussian width of its intersection with a unit Euclidean ball [4], or the closely related concept of *statistical dimension* of the cone, proposed by [1]. The aim of this appendix is to prove the upper bound on the statistical dimension $\Omega_{k,q}$ given in Proposition 1 of the main text.

## 3.1   The statistical dimension and its properties

Let us first briefly recall what the statistical dimension of a convex regularizer $\Omega : \mathbb{R}^{m_1 \times m_2} \to \mathbb{R}$ refers to, and how it is related to efficiency of the regularizer to recover a matrix $Z \in \mathbb{R}^{m_1 \times m_2}$. For that purpose, we first define the tangent cone $T_\Omega(Z)$ of $\Omega$ at $Z$ as the closure of the cone of descent directions, i.e.,

$$
T_\Omega(Z) := \overline{\bigcup_{\tau > 0} \left\{ H \in \mathbb{R}^{m_1 \times m_2} \ : \ \Omega(Z + \tau H) \leq \Omega(Z) \right\}}.
$$

The statistical dimension $\mathfrak{S}(Z, \Omega)$ of $\Omega$ at $Z$ can then be formally defined as

$$
\mathfrak{S}(Z, \Omega) := \mathbb{E}\left[ \left\| \Pi_{T_\Omega(Z)}(G) \right\|_F^2 \right],
$$

where $G$ is a random matrix with i.i.d. standard normal entries and $\Pi_{T_\Omega(Z)}(G)$ is the orthogonal projection of $G$ onto the cone $T_\Omega(Z)$. The statistical dimension is a powerful tool to quantify the statistical performance of a regularizer in various contexts, as the following non-exhaustive list of results shows.

- **Exact recovery with random measurements.** Suppose we observe $y = \mathcal{X}(Z^\star)$ where $\mathcal{X} : \mathbb{R}^{m_1 \times m_2} \to \mathbb{R}^n$ is a random linear map represented by random design matrices $X_i$ $i = 1, \ldots, n$ having iid entries drawn from $\mathcal{N}(0, 1/n)$. Then [4, Corollary 3.3] shows that

$$
\hat{Z} = \arg\min_Z \Omega(Z) \quad \text{s.th.} \quad \mathcal{X}(Z) = y \tag{13}
$$

is equal to $Z^\star$ with overwhelming probability as soon as $n \geq \mathfrak{S}(Z^\star, \Omega)$. In addition [1, Theorem II] show that a phase transition occurs at $n = \mathfrak{S}(Z^\star, \Omega)$ between a situation where recovery fails with large probability (for $n \leq \mathfrak{S}(Z^\star, \Omega) - \gamma\sqrt{m_1 m_2}$, for some $\gamma > 0$) to a situation where recovery works with large probability (for $n \geq \mathfrak{S}(Z^\star, \Omega) + \gamma\sqrt{m_1 m_2}$).

- **Robust recovery with random measurements.** Suppose we observe $y = \mathcal{X}(Z^\star) + \epsilon$ where $\mathcal{X}$ is again a random linear map, and in addition the observation is corrupted by a

random noise $\epsilon \in \mathbb{R}^n$. If the noise is bounded as $\|\epsilon\|_2 \leq \delta$, then [4, Corollary 3.3] show that

$$\hat{Z} = \arg\min_{Z} \Omega(Z) \quad \text{s.th.} \quad \|\mathcal{X}(Z) - y\|_2 \leq \delta \tag{14}$$

satisfies $\|\hat{Z} - Z^\star\|_2 \leq 2\delta/\eta$ with overwhelming probability as soon as $n \geq (\mathfrak{S}(Z^\star, \Omega) + \frac{3}{2})/(1 - \eta)^2$.

- **Denoising.** Assume a collection of noisy observations $X_i = Z^\star + \sigma\epsilon_i$ for $i = 1, \cdots, n$ is available where $\epsilon_i \in \mathbb{R}^{m_1 \times m_2}$ has i.i.d. $\mathcal{N}(0, 1)$ entries, and let $Y = \frac{1}{n} \sum_{i=1}^{n} X_i$ denote their average. [3, Proposition 4] prove that

$$\hat{Z} = \arg\min_{Z} \|Z - Y\|_F \quad \text{s.th.} \quad \Omega(Z) \leq \Omega(Z^\star) \tag{15}$$

satisfies $\mathbb{E}\|\hat{Z} - Z^\star\|_F^2 \leq \frac{\sigma^2}{n}\mathfrak{S}(Z^\star, \Omega)$.

- **Demixing.** Given two matrices $Z^\star, V^\star \in \mathbb{R}^{m_1 \times m_2}$, suppose we observe $y = \mathcal{U}(Z^\star) + V^\star$ where $\mathcal{U} : \mathbb{R}^{m_1 \times m_2} \mapsto \mathbb{R}^{m_1 \times m_2}$ is a random orthogonal operator. Given two convex functions $\Gamma, \Omega : \mathbb{R}^{m_1 \times m_2} \to \mathbb{R}$, [1, Theorem III] show that

$$(\hat{Z}, \hat{V}) = \arg\min_{(Z,V)} \Omega(Z) \quad \text{s.th.} \quad \Gamma(V) \leq \Gamma(V^\star) \quad \text{and} \quad y = \mathcal{U}(Z) + V$$

is equal to $(Z^\star, V^\star)$ with probability at least $1 - \eta$ provided that

$$\mathfrak{S}(Z^\star, \Omega) + \mathfrak{S}(V^\star, \Gamma) \leq m_1 m_2 - 4\sqrt{m_1 m_2 \log \frac{4}{\eta}}.$$

Conversely if $\mathfrak{S}(Z^\star, \Omega) + \mathfrak{S}(V^\star, \Gamma) \geq m_1 m_2 + 4\sqrt{m_1 m_2 \log \frac{4}{\eta}}$, the demixing fails with probability at least $1 - \eta$.

## 3.2 Supporting subspace, projections and subgradients

The main argument of the proof will be presented in section 3.3. It requires to construct an element of the normal cone, the cone that is polar[1] to the tangent cone, and which is also the conic hull of the subgradient. It is therefore important to characterize this normal cone, the subgradient and in particular some subspaces related to them.

At a sparse vector, a norm inducing sparsity is non-differentiable, and so the normal cone is not reduced to a single ray. However, a key property is that in general, the normal cone is not full dimensional either, but rather contained in a subspace of low codimension, say $s$. This is due to the fact that the projection of the subgradient on a subspace of dimension $s$ associated with the support of the sparse vector that we will call the *supporting subspace* is a singleton. This *supporting subspace* (although not named in general) is well-known and has been exploited for the analysis of so-called decomposable norms.

In the case of the $\ell_1$ norm the *supporting subspace* is simply the one spanned by the non-zero coordinates. In the case of the trace norm, computed at a matrix $A \in \mathbb{R}^{m_1 \times m_2}$ the supporting subspace, that we will denote $\operatorname{span}(A)$, is the range of the linear application $(L, R) \mapsto LA + AR \in \mathbb{R}^{m_1 \times m_2}$. In the case of $\Omega_{k,q}$, computed at an atom $A$ of $\mathcal{A}_{k,q}$, the supporting subspace will combine some properties of the supporting subspaces of the $\ell_1$-norm and of the trace norm, since put informally, it can be defined the range of a restriction of the linear application $(L, R) \mapsto LA + AR$ where $L$ and $R$ are restricted to have the same column and row supports as $A$.

More formally, let $\operatorname{span}(A)$ denote the subspace of $\mathbb{R}^{m_1 \times m_2}$ defined by

$$\operatorname{span}(A) = \left\{ LA + AR, L \in \mathbb{R}^{m_1 \times m_1}, R \in \mathbb{R}^{m_2 \times m_2} \right\},$$

and by $\mathcal{P}_A$ and $\mathcal{P}_A^\perp$ the orthogonal projectors onto $\operatorname{span}(A)$ and $\operatorname{span}^\perp(A)$ respectively. We have the closed-form expressions $\mathcal{P}_A^\perp(Z) = (Id_{m_1} - UU^\top)Z(Id_{m_2} - VV^\top)$ where $A = U\Sigma V^\top$ is the singular value decomposition of $A$.

Consider now the subspace
$$\text{span}_{I,J}(A) = \left\{ L_{I,I}A_{I,J} + A_{I,J}R_{J,J}, L \in \mathbb{R}^{m_1 \times m_1}, R \in \mathbb{R}^{m_2 \times m_2} \right\}$$
and its orthogonal
$$\text{span}_{I,J}^\perp(A) = \left\{ Z \in \mathbb{R}^{m_1 \times m_2} \ , \ A_{I,J}Z_{I,J}^\top = A_{I,J}^\top Z_{I,J} = 0 \right\} \ .$$

It is easy to check that the projectors $\Pi_{A,I,J}$ onto $\text{span}_{I,J}(A)$ and $\Pi_{A,I,J}^\perp$ onto $\text{span}_{I,J}^\perp(A)$ satisfy respectively

$$\Pi_{A,I,J}(Z) = \mathcal{P}_{A_{I,J}}(Z_{I,J}) \quad \text{and} \quad \Pi_{A,I,J}^\perp(Z) = Z - \Pi_{A,I,J}(Z) = Z - \mathcal{P}_{A_{I,J}}(Z_{I,J}) \ .$$

As we will see, $\text{span}_{I_0,J_0}^\perp(A)$ is needed to define the subgradient of $\Omega_{k,q}$, and is the supporting subspace for $\Omega_{k,q}$ at an element $A \in \mathcal{A}_{k,q}$.

**Lemma 5.** *If $A = ab^\top \in \mathcal{A}_{k,q}$ with $I_0 = \text{supp}(a)$ and $J_0 = \text{supp}(b)$, then the dimension of $\text{span}_{I_0,J_0}(A)$ is $k + q - 1$*

*Proof.* For $A = ab^\top$, the range of $L \mapsto L_{I_0,I_0}A_{I_0,J_0}$ equals the range of $\alpha_{I_0} \mapsto \alpha_{I_0}b^\top$ which has dimension $|I_0| = k$. By the same token, the range of $R \mapsto A_{I_0,J_0}R_{J_0,J_0}$ has dimension $q$. Using the definition of $\text{span}_{I_0,J_0}(A)$
$$\text{span}_{I_0,J_0}(A) = \left\{ \alpha_{I_0}b^\top + a\beta_{J_0}^\top, \alpha \in \mathbb{R}^{m_1}, \beta \in \mathbb{R}^{m_2} \right\}$$
therefore by the inclusion-exclusion principle $s = \dim\left(\text{span}_{I_0,J_0}(A)\right) = k + q - 1$. $\square$

The following lemma provides an explicit description of the subdifferential of $\Omega_{k,q}$ at an atom $A = ab^\top \in \mathcal{A}_{k,q}$.

**Lemma 6.** *The subdifferential of $\Omega_{k,q}$ at $A \in \mathcal{A}_{k,q}$ is*
$$\partial\Omega_{k,q}(A) = \left\{ A + Z : \ AZ_{I_0,J_0}^\top = 0, \ A^\top Z_{I_0,J_0} = 0, \ \forall (I,J) \in \mathcal{G}_k \times \mathcal{G}_q \quad \|A_{I,J} + Z_{I,J}\|_{\text{op}} \leq 1 \right\} \ .$$

*Lemma 6.* Combining the general characterization of the subgradient of a norm (see the introduction of [15], or [2] equation (1.4)) and (2), we get:
$$\partial\Omega_{k,q}(A) = \arg\max_B \{ \langle B, A \rangle \ : \ \Omega_{k,q}^*(B) \leq 1 \}$$
$$= \arg\max_B \{ \langle B, A \rangle \ : \ \|B_{I,J}\|_{\text{op}} \leq 1 \ , \ \forall I \in \mathcal{G}_k, \forall J \in \mathcal{G}_q \} \ .$$

As the dual of the dual of a norm equals the norms itself, $\max\{\langle B, A \rangle \ , \ \Omega_{k,q}^*(B) \leq 1\} = \Omega_{k,q}(A)$. Since $A$ is an atom, by construction $\Omega_{k,q}(A) = 1$. We also have $\langle A, A \rangle = 1$, so $\Omega_{k,q}(A) = \langle A, A \rangle$. On the other hand for all pairs of index sets $(I,J) \in \mathcal{G}_k \times \mathcal{G}_q$, $\|A_{I,J}\|_{\text{op}} \leq 1$. It follows that that $A$ is a subgradient of $\Omega_{k,q}$ at $A$. Letting $Z$ denote a matrix such that $A + Z \in \partial\Omega_{k,q}(A)$, the condition $\forall (I,J) \in \mathcal{G}_k \times \mathcal{G}_q$, $\|A_{I,J} + Z_{I,J}\|_{\text{op}} \leq 1$ follows. To prove $AZ_{I_0,J_0}^\top = 0$ and $A^\top Z_{I_0,J_0} = 0$, we can introduce vectors $\alpha, \beta$ such that $\mathcal{P}_A(Z_{I_0,J_0}) = \alpha b^\top + a\beta^\top$. We decompose $\beta = c_1 b + \beta'$ with $b^\top\beta' = 0$ and similarly $\alpha = c_2 a + \alpha'$ with $a^\top\alpha' = 0$. The condition $\langle A + Z, A \rangle = 1$ implies $\langle \mathcal{P}_A(Z_{I_0,J_0}), A \rangle = 0$, so $c_1 = -c_2$. Therefore $\mathcal{P}_A(Z_{I_0,J_0}) = \alpha'b^\top + a\beta'^\top$ and as a consequence

$$\|A + Z_{I_0,J_0}\|_{\text{op}} \geq \|\mathcal{P}_A(A + Z_{I_0,J_0})\|_{\text{op}}$$
$$= \|ab^\top + \alpha'b^\top + a\beta'^\top\|_{\text{op}}$$
$$\geq \left( \frac{a + \alpha'}{\|a + \alpha'\|_2} \right)^\top \left( ab^\top + \alpha'b^\top + a\beta'^\top \right) b$$
$$= \|a + \alpha'\|_2 = \sqrt{\|a\|_2^2 + \|\alpha'\|_2^2} \ .$$

Therefore $\|A + Z_{I_0,J_0}\|_{\text{op}} > 1$ unless $\alpha' = 0$ and similarly we get $\beta' = 0$. We conclude that $\mathcal{P}_A(Z_{I_0,J_0}) = 0$ or equivalently $AZ_{I_0,J_0}^\top = 0$ and $A^\top Z_{I_0,J_0} = 0$. Conversely take $B = A + Z$ where $AZ_{I_0,J_0}^\top = 0$, $A^\top Z_{I_0,J_0} = 0$, $\forall (I,J) \in \mathcal{G}_k \times \mathcal{G}_q$ $\|A_{I,J} + Z_{I,J}\|_{\text{op}} \leq 1$. It is immediate that $B$ satisfies $\Omega_{k,q}^*(B) \leq 1$ and maximizes $\langle B, A \rangle$ over the unit ball of the dual norm by taking the value 1. $\square$

### 3.3  Proof of Proposition 1 of the main text

*Proof.* To prove the result, we use the fact that the statistical dimension can be expressed as the mean quadratic Euclidean distance to the normal cone. Indeed, if $T_\Omega(A)$ denotes the tangent cone of a regularizer $\Omega$ at $A \in \mathbb{R}^p$, then the normal cone $N_\Omega(A)$ is the polar cone of $T_\Omega(A)$, or equivalently the conic hull of the subgradient $\partial\Omega(A)$ of $\Omega$ at $A$ ([13], Theorem 23.7). Then, proposition 3.6 in [4] shows that

$$\mathfrak{S}(A,\Omega) := \mathbb{E}\Big[\text{dist}(G, N_\Omega(A))^2\Big],$$

where $\text{dist}(G, N_\Omega(A))$ denotes the Euclidean distance of the Gaussian vector $G$ with i.i.d. standard normal entries to the normal cone $N_\Omega(A)$.

We therefore have $\mathfrak{S}(A,\Omega) \le \mathbb{E}\Big[\|G - \Xi(G)\|_F^2\Big]$ for any $\Xi(G) \in N_\Omega(A)$. Following [4], who prove a couple of results using this technique, we construct a matrix $\Xi(G)$ belonging to the normal cone, for which this squared distance be sharply upper bounded.

Let $A = ab^\top$ be an element of $\mathcal{A}_{k,q}$, with $I_0 = \text{supp}(a)$ and $J_0 = \text{supp}(b)$. Let $u_I = \frac{a_I}{\|a_I\|}$ and $v_J = \frac{b_J}{\|b_J\|}$. Note that while $a_I$ is a subvector of $a$, the notation $u_I$ does not refer to a subvector of some vector $u$ and that therefore $[u_I]_{I_0} \ne [u_{I_0}]_I = a_I$ since $\|a_{I_0}\| = \|a\| = 1$. We will use $i = |I \backslash I_0|$ and $j = |J \backslash J_0|$.

Define $\Xi(G) = \epsilon(G)\, ab^\top + \tilde{G}$, with $\tilde{G} = \Pi_{A,I_0,J_0}^\perp(G)$ and let $\epsilon(G)^2$ be equal to

$$\frac{16}{\gamma^2}\|G_{I_0,J_0}\|_{\text{op}}^2 \vee \max_{\substack{I \in \mathcal{G}_k \\ J \in \mathcal{G}_q}}\|G_{IJ}\|_{\text{op}}^2 \vee \max_{\substack{0 \le i < k \\ 0 \le j < q \\ (i,j)\neq(0,0)}} \frac{8}{\gamma\left(\frac{i}{k}+\frac{j}{q}\right)} \max_{\substack{|I \backslash I_0|=i \\ |J \backslash J_0|=j}} \|G_{I \cap I_0, J \backslash J_0}^\top u_I\|_2^2 + \|G_{I \backslash I_0, J \cap J_0} v_J\|_2^2,$$

where, for short, we wrote $\gamma = \gamma(a,b)$ for the signal strength of $(a,b)$ (see Definition 3 in the main text).

We prove in Section 3.3.2 (see the main lemma 9) that this $\Xi(G)$ satisfies $\Xi(G) \in N_{\Omega_{k,q}}(A)$.

Using the decomposition $G = \Pi_{A,I_0,J_0}(G) + \Pi_{A,I_0,J_0}^\perp(G)$ and the fact that $\Pi_{A,I_0,J_0}(G)$ is by construction orthogonal to the normal cone, we have

$$\mathfrak{S}(A,\Omega_{k,q}) \le \mathbb{E}\|G - \Xi(G)\|_F^2 = \mathbb{E}\|\epsilon(G)ab^\top - \Pi_{A,I_0,J_0}(G)\|_F^2$$

$$= \mathbb{E}\|\epsilon(G)ab^\top\|_F^2 + \|\Pi_{A,I_0,J_0}(G)\|_F^2 \tag{16}$$

$$= \mathbb{E}\,\epsilon(G)^2 + (k+q-1), \tag{17}$$

where (16) follows from Pythagoras theorem and (17) is due to $\|ab^\top\|_F = 1$ and the fact that $\|\Pi_{A,I_0,J_0}(G)\|_F^2$ follows a chi-square distribution with $k + q - 1$ degrees of freedom, since by lemma 5 this is the dimension of $\text{span}_{I_0,J_0}(A)$.

The rest of the proof consists in bounding the three terms of $\mathbb{E}\,\epsilon(G)^2$. By lemmata 7 and 4 we respectively have:

$$\frac{16}{\gamma^2}\,\mathbb{E}[\|G_{I_0,J_0}\|_{\text{op}}^2] \le \frac{64}{\gamma^2}(k+q+1),$$

$$\mathbb{E}\max_{I,J}\|G_{I,J}\|_{\text{op}}^2 \le 16\left[\left(k\log\frac{m_1}{k} + q\log\frac{m_2}{q}\right) + 2(k+q)\right].$$

The third term is bounded in Lemma 8 by $\frac{48}{\gamma}(k \vee q)\log\left((m_1-k)\vee(m_2-q)\right) + \frac{64}{\gamma}(k \vee q)$.

Combining these terms with equation (17), we obtain

$$\mathfrak{S}(A,\Omega_{k,q}) \le \left(\frac{64}{\gamma^2} + \frac{64}{\gamma} + 35\right)(k+q+1) + 16\left(k\log\frac{m_1}{k} + q\log\frac{m_2}{q}\right) + \frac{48}{\gamma}(k \vee q)\log(m_1 \vee m_2).$$

$\square$

### 3.3.1 Upper bounds for $\epsilon(G)^2$

This section provides upper bounds on the terms that compose $\mathbb{E}\epsilon(G)^2$.

**Lemma 7.** *For every matrix $G \in \mathbb{R}^{m_1 \times m_2}$ with entries drawn iid from $\mathcal{N}(0,1)$, and for $I_0 \in \mathcal{G}_k$, $J_0 \in \mathcal{G}_q$ we have*

$$\mathbb{E}[\|G_{I_0,J_0}\|_{\text{op}}^2] \leq 4(k+q) + 4 \tag{18}$$

*Lemma 7.* From (8), we have for $s \geq 0$,

$$\mathbb{P}(\|G_{I_0,J_0}\|_{\text{op}} > \sqrt{k} + \sqrt{q} + s) \leq \exp(-s^2/2)$$

and as a consequence, and given that $\left(\sqrt{k} + \sqrt{q} + s\right)^2 \leq 2\left((\sqrt{k} + \sqrt{q})^2 + s^2\right)$,

$$\mathbb{P}\left[\|G_{I_0,J_0}\|_{\text{op}}^2 > 2\left((\sqrt{k} + \sqrt{q})^2 + s^2\right)\right] \leq \exp(-s^2/2).$$

Setting $t = 2s^2$ yields $\quad \mathbb{P}(\|G_{I_0,J_0}\|_{\text{op}}^2 > 4(k+q) + t) \leq \exp(-t/4)$. It follows that

$$
\begin{aligned}
\mathbb{E}[\|G_{I_0,J_0}\|_{\text{op}}^2] &= \int_0^\infty \mathbb{P}(\|G_{I_0,J_0}\|_{\text{op}}^2 \geq t')dt' \\
&= \int_0^{4(k+q)} dt' + \int_{4(k+q)}^\infty \mathbb{P}(\|G_{I_0,J_0}\|_{\text{op}}^2 \geq t')dt' \\
&\leq 4(k+q) + \int_0^\infty \exp(-t/4)dt = 4(k+q) + 4.
\end{aligned}
$$

$\square$

**Lemma 8.**

$$\mathbb{E}\max_{i,j} \frac{8}{\gamma\left(\frac{i}{k} + \frac{j}{q}\right)} \max_{\substack{|J\backslash J_0|=j \\ |I\backslash I_0|=i}} \|G_{I\cap I_0, J\backslash J_0}^\top u_I\|_2^2 + \|G_{I\backslash I_0, J\cap J_0} v_J\|_2^2$$

$$\leq \frac{48}{\gamma}(k \vee q)\log\left((m_1 - k) \vee (m_2 - q)\right) + \frac{64}{\gamma}(k \vee q).$$

*Lemma 8.* As the sets $I \cap I_0 \times J\backslash J_0$ and $I\backslash I_0 \times J \cap J_0$ are disjoint, and $u_I, v_J$ of unit length, the random variable

$$M_{I,J} = \|G_{I\cap I_0, J\backslash J_0}^\top u_I\|_2^2 + \|G_{I\backslash I_0, J\cap J_0} v_J\|_2^2$$

follows a chi-squared distribution with $i + j$ degrees of freedom: $M_{I,J} \sim \chi_{i+j}^2$, where $i = |I\backslash I_0|$ and $j = |J\backslash J_0|$. Using Chernoff's inequality and the form of the chi-square moment generating function, we have that for any fixed real number $\alpha$ and fixed index sets $I$ and $J$, for all $t \in (0, 1/2)$,

$$\mathbb{P}\left[M_{I,J} > \alpha\right] = \mathbb{P}\left[e^{tM_{I,J}} > e^{t\alpha}\right] \leq e^{-t\alpha}\,\mathbb{E}\,e^{tM_{I,J}} = e^{-t\alpha}(1-2t)^{-\frac{i+j}{2}}.$$

So, taking the maximum over index sets $I$ and $J$ with the same intersection sizes with $I_0$ and $J_0$ respectively, and using a union bound on the independent choices of $I$ and $J$,

$$
\begin{aligned}
\mathbb{P}\left[\max_{\substack{|I\backslash I_0|=i \\ |J\backslash J_0|=j}} M_{I,J} > \alpha\right] &\leq \binom{m_1 - k}{i}\binom{m_2 - q}{j}\exp\left\{-t\alpha - \frac{i+j}{2}\log(1-2t)\right\} \\
&\leq \exp\left\{-t\alpha - \frac{i+j}{2}\log(1-2t) + i\log(m_1 - k) + j\log(m_2 - q)\right\}.
\end{aligned}
$$

Take $\alpha = \lambda(i + j)$, we have for any $t < 1/2$, assuming w.l.o.g. $m_1 - k \geq m_2 - q$,

$$
\begin{aligned}
\mathbb{P}\left[\max_{\substack{|I\backslash I_0|=i \\ |J\backslash J_0|=j}} M_{I,J} > \lambda(i+j)\right] &\leq \exp\left\{-t\lambda(i+j) - \frac{i+j}{2}\log(1-2t) + i\log(m_1 - k) + j\log(m_2 - q)\right\} \\
&\leq \exp\left\{(i+j)\left(-t\lambda - \frac{1}{2}\log(1-2t) + \log(m_1 - k)\right)\right\}.
\end{aligned}
$$

Introduce $\mathcal{M}_{i,j} = \frac{1}{i+j} \max_{\substack{|I\setminus I_0|=i \\ |J\setminus J_0|=j}} M_{I,J}$, take $t = \frac{1}{2}\left(1 - \frac{1}{m_1-k}\right) < \frac{1}{2}$. Then

$$\mathbb{P}\left[\max_{\substack{0\le i<k \\ 0\le j<q \\ (i,j)\ne(0,0)}} \mathcal{M}_{i,j} > \lambda\right] \le \sum_{\substack{0\le i<k \\ 0\le j<q \\ (i,j)\ne(0,0)}} \exp\left\{(i+j)\left(-\frac{1}{2}\left(1 - \frac{1}{m_1-k}\right)\lambda + \frac{3}{2}\log(m_1-k)\right)\right\}$$

$$= \sum_{i=0}^{k-1}\beta^i \sum_{j=0}^{q-1}\beta^j - 1 = \frac{1-\beta^k}{1-\beta}\frac{1-\beta^q}{1-\beta} - 1 \le 2\beta,$$

where $\beta = \exp\left\{-\frac{1}{2}\left(1 - \frac{1}{m_1-k}\right)\lambda + \frac{3}{2}\log(m_1-k)\right\}$.

As a consequence, we have

$$\mathbb{E}[\max_{i,j}\mathcal{M}_{i,j}] = \int_0^\infty \mathbb{P}[\max_{i,j}\mathcal{M}_{i,j} > \lambda]d\lambda$$

$$\le \int_0^{\frac{3(m_1-k)}{m_1-k-1}\log k} d\lambda + 2\int_{\frac{3(m_1-k)}{m_1-k-1}\log(m_1-k)}^\infty \exp\left\{\frac{3}{2}\log(m_1-k) - \frac{1}{2}\left(1 - \frac{1}{m_1-k}\right)\lambda\right\}d\lambda$$

$$\le \frac{3(m_1-k)}{m_1-k-1}\log k + 4\frac{m_1-k}{m_1-k-1}$$

$$\le 6\log(m_1-k) + 8 .$$

It follows that

$$\mathbb{E} \max_{\substack{0\le i<k \\ 0\le j<q \\ (i,j)\ne(0,0)}} \frac{8}{\gamma\left(\frac{i}{k}+\frac{j}{q}\right)} \max_{\substack{|J\setminus J_0|=j \\ |I\setminus I_0|=i}} \|G_{I\cap I_0, J\setminus J_0}^\top u_I\|_2^2 + \|G_{I\setminus I_0, J\cap J_0} v_J\|_2^2$$

$$\le \frac{48}{\gamma}(k\vee q)\log\left((m_1-k)\vee(m_2-q)\right) + \frac{64}{\gamma}(k\vee q) . \qquad (19)$$

$\square$

### 3.3.2  The scaling factor $\epsilon(G)$ ensures that $\Xi(G) \in N_{\Omega_{k,q}}(A)$

We will use the notation $u_I = \frac{a_I}{\|a_I\|}$ and $v_J = \frac{b_J}{\|b_J\|}$. We will use several times the fact that $a = a_{I_0}$, $a_I = a_{I\cap I_0}$ and the corresponding properties for $b$, $u$ and $v$.

The objective of this appendix is to prove the following lemma:

**Lemma 9.** *Let $\epsilon(G)^2$ be equal to*

$$\frac{16}{\gamma^2}\|G_{I_0,J_0}\|_{op}^2 \vee \max_{\substack{I\in\mathcal{G}_k \\ J\in\mathcal{G}_q}}\|G_{IJ}\|_{op}^2 \vee \max_{\substack{0\le i<k \\ 0\le j<q \\ (i,j)\ne(0,0)}} \frac{8}{\gamma\left(\frac{i}{k}+\frac{j}{q}\right)} \max_{\substack{|I\setminus I_0|=i \\ |J\setminus J_0|=j}} \|G_{I\cap I_0, J\setminus J_0}^\top u_I\|_2^2 + \|G_{I\setminus I_0, J\cap J_0} v_J\|_2^2.$$

*Then, for every $G \in \mathbb{R}^{m_1\times m_2}$, the matrix $\Xi(G) = \epsilon(G)A + \Pi_{A,I_0,J_0}^\perp(G)$ belongs to the normal cone of $\Omega_{k,q}$ at $A$.*

*Lemma 9.* Recall that the subgradient of the norm $\Omega$ at $A$ is defined as

$$\partial\Omega(A) = \{M \in \mathbb{R}^{m_1\times m_2} \mid \langle A, M\rangle_F = \Omega(A),\ \Omega^*(M) \le 1\},$$

and that the normal cone $N_\Omega(A)$ is the conic hull of the subgradient. Given that $\langle ab^\top, \Xi(G)\rangle = \epsilon(G) = \epsilon(G)\Omega_{k,q}(ab^\top)$, we have that $\Xi(G)$ is in $N_{\Omega_{k,q}}(ab^\top)$ if and only if $\epsilon(G)^{-1}\Xi(G)$ belongs to the subgradient $\partial\Omega(ab^\top)$. But given that the property $\langle ab^\top, \epsilon(G)^{-1}\Xi(G)\rangle = \Omega_{k,q}(ab^\top)$ is satisfied, it is sufficient to show that $\Omega_{k,q}^*\big(\epsilon(G)^{-1}\Xi(G)\big) \le 1$.

We therefore need to prove that, for all $(I,J)$, we have $\|A_{IJ} + \epsilon(G)^{-1}\tilde{G}_{IJ}\|_{op} \le 1$. This is equivalent to requiring that

$$\|A_{IJ}^\top + \epsilon(G)^{-1}\Pi_{A,I,J}(\tilde{G})\|_{op} \le 1 \qquad \text{and} \qquad \epsilon(G)^{-1}\|\mathcal{P}_A^\perp(\tilde{G}_{I,J})\|_{op} \le 1. \qquad (20)$$

First the second inequality of (20) is satisfied since

$$\|\mathcal{P}_A^\perp(\tilde{G}_{I,J})\|_{\mathrm{op}} \leq \|\tilde{G}_{I,J}\|_{\mathrm{op}} = \big\|[\Pi_{A,I_0,J_0}^\perp(G)]_{IJ}\big\|_{\mathrm{op}} \leq \big\|[G]_{IJ}\big\|_{\mathrm{op}} \leq \epsilon(G)^2.$$

There thus remains to prove the first inequality of (20). Note that the matrix $A_{IJ}^\top + \epsilon(G)^{-1}\Pi_{A,I,J}(\tilde{G})$ has rank 2, so its Frobenius norm by at most a factor of $\sqrt{2}$ larger than its operator norm. Working with the Frobenius norm is more convenient, so knowing that

$$\|A_{IJ} + \epsilon(G)^{-1}\Pi_{A,I,J}(\tilde{G})\|_{\mathrm{op}} \leq \|A_{IJ}^\top + \epsilon(G)^{-1}\Pi_{A,I,J}(\tilde{G})\|_F \ ,$$

we will establish an upper bound on the latter quantity.

By definition

$$\Pi_{A,I,J}(\tilde{G}) = u_I u_I^\top \tilde{G} + \tilde{G}v_J v_J^\top + u_I^\top \tilde{G}v_J\, u_I v_J^\top,$$

and by lemma 10, denoting $\nu_{I,J}(G) = \|A_{IJ} + \epsilon(G)^{-1}\Pi_{A,I,J}(\tilde{G}_{IJ})\|_F^2$ we then have

$$\nu_{I,J}(G) = \big\| \|a_I\|\|b_J\|u_I v_J^\top + \epsilon(G)^{-1}(u_I u_I^\top \tilde{G}_{IJ} + \tilde{G}_{IJ}v_J v_J^\top - u^\top \tilde{G}_{IJ}v_J\, u_I v_J^\top)\big\|_F^2$$

$$\leq \|a_I\|^2\|b_J\|^2 + 2\,\|a_I\|\|b_J\|\frac{1}{\epsilon(G)}\,u_I^\top \tilde{G}_{IJ}v_J + \frac{1}{\epsilon(G)^2}(u_I^\top \tilde{G}_{IJ}\tilde{G}_{IJ}^\top u_I + v_J^\top \tilde{G}_{IJ}^\top \tilde{G}_{IJ}v_J)\,.$$

Now, recall that $\tilde{G} = \Pi_{A,I_0,J_0}^\perp(G)$. Lemma 11 explicits the structure of $\tilde{G}$. It is exploited in 12 to obtain the inequalities of lemma 13 that yield

$$\nu_{I,J}(G) \leq \ \|a_I\|^2\|b_J\|^2 + \frac{2}{\epsilon(G)}\,\|a_I\|\|b_J\|\,\|a_{I_0\setminus I}\|\,\|b_{J_0\setminus J}\|\,\|G_{I_0 J_0}\|_{\mathrm{op}}$$

$$+ \frac{1}{\epsilon(G)^2}(\|G_{I\cap I_0,J\setminus J_0}^\top u_I\|_2^2 + 2\,\|a_{I_0\setminus I}\|^2\,\|G_{I_0,J_0}\|_{\mathrm{op}}^2)$$

$$+ \frac{1}{\epsilon(G)^2}(\|G_{I\setminus I_0,J\cap J_0}v_J\|_2^2 + 2\,\|b_{J_0\setminus J}\|^2\,\|G_{I_0,J_0}\|_{\mathrm{op}}^2). \qquad (21)$$

Finally, using the fact that $\epsilon(G)^2$ equals

$$\frac{16}{\gamma^2}\|G_{I_0,J_0}\|_{\mathrm{op}}^2 \vee \max_{\substack{I\in\mathcal{G}_k \\ J\in\mathcal{G}_q}}\|G_{IJ}\|_{\mathrm{op}}^2 \vee \max_{\substack{0\leq i<k \\ 0\leq j<q \\ (i,j)\neq(0,0)}}\frac{8}{\gamma\left(\frac{i}{k}+\frac{j}{q}\right)}\max_{\substack{|I\setminus I_0|=i \\ |J\setminus J_0|=j}}\|G_{I\cap I_0,J\setminus J_0}^\top u_I\|_2^2 + \|G_{I\setminus I_0,J\cap J_0}v_J\|_2^2,$$

and given that inequality (21) implies the inequality

$$\nu_{I,J}(G) \leq \|a_I\|^2\|b_J\|^2 + \frac{\gamma}{2}\,\|a_I\|\|b_J\|\|a_{I_0\setminus I}\|\|b_{J_0\setminus J}\| + \frac{\gamma}{8}\left(\frac{i}{k}+\frac{j}{q}\right) + \frac{\gamma^2}{8}\left(\|a_{I_0\setminus I}\|^2 + \|b_{J_0\setminus J}\|^2\right)\,.$$

Define $\alpha := \|a_{I_0\setminus I}\|^2 = 1 - \|a_I\|^2$ and $\beta := \|b_{J_0\setminus J}\|^2 = 1 - \|b_J\|^2$.

With these notations and rearranging the terms, we can rewrite the above inequality as

$$\nu_{I,J}(G) \leq (1-\alpha)(1-\beta) + \frac{\gamma}{2}\sqrt{\alpha\beta(1-\alpha)(1-\beta)} + \frac{\gamma^2}{8}(\alpha+\beta) + \frac{\gamma}{8}\left(\frac{i}{k}+\frac{j}{q}\right)\,.$$

Since $0 \leq \alpha,\beta \leq 1$ and using $\sqrt{\alpha\beta} \leq \frac{1}{2}(\alpha+\beta)$, we have

$$\alpha\beta \leq \frac{1}{2}(\alpha+\beta) \quad \text{and} \quad \sqrt{\alpha\beta(1-\alpha)(1-\beta)} \leq \frac{1}{2}(\alpha+\beta).$$

These inequalities imply that

$$\nu_{I,J}(G) \leq 1 + (\alpha+\beta)\left(-1+\frac{1}{2}+\frac{\gamma}{4}+\frac{\gamma^2}{8}\right) + \frac{\gamma}{8}\left(\frac{i}{k}+\frac{j}{q}\right)$$

By definition of $\gamma = \min_{\substack{\iota \in I_0 \\ \iota' \in J_0}} \left( k\, a_\iota^2, q\, b_{\iota'}^2 \right)$, we have $\frac{i}{k} \leq \frac{\alpha}{\gamma}$ and $\frac{j}{q} \leq \frac{\beta}{\gamma}$. Moreover, given that $0 \leq \gamma \leq 1$, we have $\frac{4}{\gamma} - 2 - \gamma = \frac{1}{\gamma}(4 - 2\gamma - \gamma^2) \geq \frac{1}{\gamma}$, so that factorizing $\frac{\gamma}{8}$ in the previous expression, we obtain

$$
\begin{aligned}
\nu_{I,J}(G) &\leq 1 + \frac{\gamma}{8} \left[ \left( -\frac{4}{\gamma} + 2 + \gamma \right)(\alpha + \beta) + \left( \frac{i}{k} + \frac{j}{q} \right) \right] \\
&\leq 1 + \frac{\gamma}{8} \left[ -\frac{1}{\gamma}(\alpha + \beta) + \left( \frac{i}{k} + \frac{j}{q} \right) \right] \\
&\leq 1
\end{aligned}
$$

which concludes the proof.

$\square$

### 3.3.3 Technical lemmas

This section gathers the lemmata needed to prove lemma 9.

**Lemma 10.**
$$
\begin{aligned}
\|\alpha u v^\top + u u^\top G + G v v^\top - u^\top G v\, u v^\top\|_F^2 &= \alpha^2 + 2\alpha u^\top G v + u^\top G G^\top u + v^\top G^\top G v - (u^\top G v)^2 \\
&\leq \alpha^2 + 2\alpha u^\top G v + u^\top G G^\top u + v^\top G^\top G v .
\end{aligned}
$$

This lemma is proved by straightforward algebra.

**Lemma 11.** *The matrix* $\tilde{G}_{IJ} = [\Pi_{X,I_0,J_0}^\perp(G)]_{IJ}$ *is of the form* $\tilde{G}_{IJ} = \tilde{G}_1 + \tilde{G}_2$ *with*

$$
\tilde{G}_1 = G_{IJ} - G_{I \cap I_0, J \cap J_0} \quad \text{and} \quad \tilde{G}_2 = (\mathrm{Id}_I - a_I a^\top)\, G_{I_0 J_0}\, (\mathrm{Id}_J - b b_J^\top).
$$

*Lemma 11.*
$$
\begin{aligned}
\Pi_{X,I_0,J_0}^\perp(G) &= G - \Pi_{X,I_0,J_0}(G) \\
&= G - a_{I_0} a_{I_0}^\top G_{I_0 J_0} - G_{I_0 J_0} b_{J_0} b_{J_0}^\top + a_{I_0} a_{I_0}^\top G_{I_0 J_0} b_{J_0} b_{J_0}^\top \\
&= G - G_{I_0 J_0} + (\mathrm{Id}_{I_0} - a_{I_0} a_{I_0}^\top)\, G_{I_0 J_0}\, (\mathrm{Id}_{J_0} - b_{J_0} b_{J_0}^\top),
\end{aligned}
$$
so that $\quad [\Pi_{X,I_0,J_0}^\perp(G)]_{IJ} = G_{IJ} - G_{I \cap I_0, J \cap J_0} + (\mathrm{Id}_I - a_I a^\top)\, G_{I_0 J_0}\, (\mathrm{Id}_J - b b_J^\top).$

$\square$

**Lemma 12.** *We have* $\quad u_I^\top \tilde{G}_1 = u_I^\top G_{I \cap I_0, J \setminus J_0} \quad$ *and* $\quad \tilde{G}_1 v_J = G_{I \setminus I_0, J \cap J_0} v_J .$

*Lemma 12.* Given that $\mathrm{supp}(u_I) \subset I_0$, we have
$$
u_I^\top \tilde{G}_1 = u_I^\top (G_{IJ} - G_{I \cap I_0, J \cap J_0}) = u_I^\top (G_{I \cap I_0, J} - G_{I \cap I_0, J \cap J_0}) = u_I^\top G_{I \cap I_0, J \setminus J_0} ,
$$
which proves the first equality. The second one is proved similarly. $\quad\square$

**Lemma 13.** *We have*
$$
\begin{aligned}
u_I^\top \tilde{G}_{IJ} v_J &\leq \|a_{I_0 \setminus I}\| \, \|b_{J_0 \setminus J}\| \, \|G_{I_0 J_0}\|_{\mathrm{op}}, \\
u_I^\top \tilde{G}_{IJ} \tilde{G}_{IJ}^\top u_I &\leq \|G_{I \cap I_0, J \setminus J_0}^\top u_I\|_2^2 + 2 \|a_{I_0 \setminus I}\|^2 \, \|G_{I_0, J_0}\|_{\mathrm{op}}^2, \\
v_J^\top \tilde{G}_{IJ}^\top \tilde{G}_{IJ} v_J &\leq \|G_{I \setminus I_0, J \cap J_0} v_J\|_2^2 + 2 \|b_{J_0 \setminus J}\|^2 \, \|G_{I_0, J_0}\|_{\mathrm{op}}^2.
\end{aligned}
$$

*Lemma 13.* Given that $\tilde{G}_{IJ} = \tilde{G}_1 + \tilde{G}_2$ and $u_I^\top \tilde{G}_1 = u_I \tilde{G}_{I \cap I_0, J \setminus J_0}$, we have $u_I^\top \tilde{G}_1 v_J = u_I^\top \tilde{G}_1 v_{J \cap J_0} = 0$, so that

$$
\begin{aligned}
u_I^\top \tilde{G}_{IJ} v_J &= u_I^\top \tilde{G}_2 v_J \\
&= u_I^\top (\mathrm{Id}_I - a_I a^\top)\, G_{I_0 J_0}\, (\mathrm{Id}_J - b b_J^\top) v_J \\
&\leq \|u_I - \|a_I\|\, a\| \, \|G_{I_0 J_0}\|_{\mathrm{op}} \, \|v_J - \|b_J\|\, b\| \\
&\leq \|a_{I_0 \setminus I}\| \, \|b_{J_0 \setminus J}\| \, \|G_{I_0 J_0}\|_{\mathrm{op}},
\end{aligned}
$$

because $\|u_I^\top (\mathrm{Id}_I - a_I a^\top)\|^2 = \big\|u_I - \|a_I\| a\big\|^2 = 1 - 2\|a_I\|^2 + \|a_I\|^2 = \|a_{I_0 \setminus I}\|^2$, and symmetrically $\big\|v_J - \|b_J\| b\big\| = \|b_{J_0 \setminus J}\|$. This shows the first inequality.

For the two next inequalities, note that

$$u_I^\top \tilde{G}_{IJ} \tilde{G}_{IJ}^\top u_I = \|\tilde{G}_{IJ}^\top u_I\|^2 = \|\tilde{G}_1^\top u_I\|^2 + \|\tilde{G}_2^\top u_I\|^2$$

because $\langle \tilde{G}_1^\top u_I, \tilde{G}_2^\top u_I \rangle = 0$ as a result of the fact that by lemma 12, $\tilde{G}_1^\top u_I$ and $\tilde{G}_2^\top u_I$ have disjoint supports.

Now $\|\tilde{G}_1^\top u_I\|^2 = \|G_{I \cap I_0, J \setminus J_0}^\top u_I\|_2^2$ and $\|\tilde{G}_2^\top u_I\| \leq 2 \|a_{I_0 \setminus I}\|^2 \|G_{I_0, J_0}\|_{\mathrm{op}}^2$, because $\|\mathrm{Id} - b_J b^\top\|_{\mathrm{op}}^2 \leq 2$ (see Lemma 14 for a proof). This shows the second inequality and the third follows by symmetry. $\qquad\square$

**Lemma 14.** $\qquad \|\mathrm{Id} - b_J b^\top\|_{\mathrm{op}}^2 \leq \dfrac{4}{3}$

*Lemma 14.* The largest singular value is attained on the span of $b_J$ and $b_{J^c}$ both on the left and on the right. Given that $\|b\| = 1$, it is therefore also the largest eigenvalue of the matrix of the linear operator restricted to this span which is equal to

$$\begin{bmatrix} (1-x) & -\sqrt{(1-x)x} \\ 0 & 1 \end{bmatrix},$$

for $x = \|b_J\|^2$. Tedious but simple calculations show that the squared operator norm of this matrix is equal to $1 - x/2 + 1/2\sqrt{x(4-3x)}$, which takes its maximum value $4/3$ for $x = 1/3$. $\qquad\square$

# 4 Lower bounds

[11] has shown that the combination $\Gamma_\mu$ of the $\ell_1$ norm and of the trace norm does not improve rates up to constants over the best of the two norms. More precisely, we can derive from [11, Theorem 3.2] the following result

**Proposition 2.** *There exists $M > 0$ and $C > 0$ such that for any $m_1, m_2, k, q \geq M$ with $m_1/k \geq M$ and $m_2/q \geq M$, for any $A \in \mathcal{A}_{k,q}$ and for any $\mu \in [0,1]$, the following holds:*

$$\mathfrak{S}(A, \Gamma_\mu) \geq C\,\zeta(a,b)\big((kq) \wedge (m_1 + m_2 - 1)\big) - 2\,,$$

*with*

$$\zeta(a,b) = 1 - \left(1 - \frac{\|a\|_1^2}{k}\right)\left(1 - \frac{\|b\|_1^2}{q}\right).$$

Note that $\zeta(a,b) \leq 1$ with equality if either $a \in \widetilde{\mathcal{A}}_k^{m_1}$ or $b \in \widetilde{\mathcal{A}}_q^{m_2}$, so in particular $\zeta(a,b) = 1$ for $ab^\top \in \widetilde{\mathcal{A}}_{k,q}$. In that case, we see that, as stated by [11], $\Gamma_\mu$ does not bring any improvement over the $\ell_1$ and trace norms taken imdividually, and in particular has a worse statistical dimension than $\Omega_{k,q}$ and $\widetilde{\Omega}_{k,q}$.

*Proof of Proposition 2.* Proposition 2 is a consequence of the following result:

**Lemma 15.** *Let $ab^\top \in \mathcal{A}_{k,q}$, $\mathcal{X} : \mathbb{R}^{m_1 \times m_2} \to \mathbb{R}^n$ a linear map from the standard Gaussian ensemble and $y = \mathcal{X}(ab^\top)$. If $n \leq \frac{1}{9} m_1 m_2$ and further*

$$n \leq n_0 := \zeta(a,b)\frac{1}{6^4}\big((kq) \wedge (m_1 + m_2 - 1)\big) - 2, \quad \text{with} \quad \zeta(a,b) = 1 - \left(1 - \frac{\|a\|_1^2}{k}\right)\left(1 - \frac{\|b\|_1^2}{q}\right),$$

*then, with probability $1 - c_1 \exp(-c_2 n_0)$, solving formulation (13) with the norm $\Gamma_\mu$ fails to recover $ab^\top$ simultaneously for any values of $\mu \in [0,1]$, where $c_1$ and $c_2$ are universal constants.*

Indeed, take $M$ such that when $m_1, m_2, k, q, m1/k, m_2/q \geq M$ then $n_0$ is large enough to ensure $1 - c_1 \exp(-c_2 n_0) > 4 \exp(-32/17)$. Then, according to Lemma 15, solving (13) with the norm $\Gamma_\mu$ fails to recover $A = ab^\top$ with probability at least $4 \exp(-32/17)$. On the other hand, [1, Theorem 7.1] shows that, when $n \geq \mathfrak{S}(A, \Gamma_\mu) + \lambda$, for any $\lambda \geq 0$, then solving (13) with the norm $\Gamma_\mu$ correctly recovers $A$ with probability at least

$$4 \exp\left(\frac{-\lambda^2/8}{\omega^2(A, \Gamma_\mu) + \lambda}\right), \tag{22}$$

where $\omega^2(A, \Gamma_\mu) = \mathfrak{S}(A, \Gamma_\mu) \wedge (m_1 m_2 - \mathfrak{S}(A, \Gamma_\mu))$. Take $\lambda = 16\omega(A, \Gamma_\mu)$, then using the fact that $\omega(A, \Gamma_\mu) \geq 1$ we get that the probability (22) is smaller than $4 \exp(-32/17)$. This implies that

$$n_0 \leq \mathfrak{S}(A, \Gamma_\mu) + \lambda \leq \mathfrak{S}(A, \Gamma_\mu) + 16\sqrt{\mathfrak{S}(A, \Gamma_\mu)} \leq 17\mathfrak{S}(A, \Gamma_\mu).$$

$\square$

*Proof of Lemma 15.* The proof consists in applying theorem 3.2 in [11] for the combination of the $\ell_1$-norm with the trace norm. We adapt slightly the notations of that paper to reflect the fact that we are working with matrices. Since we consider conic combinations of the $\ell_1$ and trace norms, the number of norms is therefore $\tau = 2$. To apply the theorem we need to specify $\kappa, \theta, d_{\min}, \gamma$ and $\mathcal{C}^\circ$ in the notations of that paper.

For each decomposable norm $\nu_j$ for $j \in \{1, 2\}$, with $\nu_1$ the $\ell_1$-norm and and $\nu_2$ the trace norm, given a point $ab^\top$ (which corresponds to the point $\mathbf{x}_0$ in [11]), the authors define

- $T_j$ the supporting subspaces and $E_j$ ($\mathbf{e}_j$ in the paper), the orthogonal projection of any subgradient of the norm in $ab^\top$ (Definition 2.1),

- $L_j$ the Lipschitz constant of $\nu_j$ with respect to the Euclidean norm (Definition 2.2),

- $\kappa_j = \dfrac{\|E_j\|_F^2}{L_j^2} \dfrac{m_1 m_2}{\dim(T_j)}$ (Definition 2.2).

Let $ab^\top \in \mathcal{A}_{k,q}$ with support $I_0 \times J_0$ and $s_a = \text{sign}(a)$, $s_b = \text{sign}(b)$. Denoting $e_{ij}$ the element of the canonical basis of $\mathbb{R}^{m_1 \times m_2}$, we have

- $T_1 = \text{span}(\{e_{ij}\}_{(i,j) \in I_0 \times J_0})$ so that $\dim(T_1) = kq$,

- $T_2 = \{av^\top + ub^\top \mid u \in \mathbb{R}^{m_1}, v \in \mathbb{R}^{m_2}\}$ so that $\dim(T_2) = m_1 + m_2 - 1$.

By definition $d_{\min} = \dim(T_1) \wedge \dim(T_2)$. We have

$$E_1 = s_a s_b^\top, \quad \|E_1\|_F^2 = kq, \quad E_2 = ab^\top, \quad \|E_2\|_F^2 = 1, \quad L_1 = \sqrt{kq}, \quad L_2 = \sqrt{m_1 \wedge m_2},$$

$$\text{and thus} \quad \kappa_1 = \frac{m_1 m_2}{kq}, \quad \kappa_2 = \frac{m_1 m_2}{(m_1 \wedge m_2)(m_1 + m_2 - 1)}, \quad \text{so that } \kappa = \kappa_1 \wedge \kappa_2 \geq \frac{1}{2}.$$

We then have $\theta$ defined as $\theta = \theta_1 \wedge \theta_2$ with $\theta_j = \|E_{\cap,j}\|_2/\|E_j\|_2$ where $E_{\cap,j}$ is the projection of $E_j$ on $T_1 \cap T_2$. But $E_2 \in T_1$ so that $\theta_2 = 1$. The situation is less simple for $E_1$. Indeed, $E_{\cap,1} = \|a\|_1 a s_b^\top + \|b\|_1 s_a b^\top - ab^\top \|a\|_1 \|b\|_1$. Some calculations lead to

$$\theta_1^2 = \frac{\|a\|_1^2}{k} + \frac{\|b\|_1^2}{q} - \frac{\|a\|_1^2}{k} \frac{\|b\|_1^2}{q},$$

hence the definition of $\zeta(a, b) = \theta^2 = \theta_1^2 \wedge \theta_2^2$. Theorem 3.2 in [11] offers the possibility of constraining the estimator to lie in a cone $\mathcal{C}$. In our case, $\mathcal{C} = \mathbb{R}^{m_1 \times m_2}$, given the definition of $\gamma$ we therefore have $\gamma \leq 2$. The result follows from applying the theorem with $\theta^2 = \zeta(a, b)$ and using $\frac{\kappa}{81\gamma^2\tau} \geq \frac{1/2}{3^4.2^2.2} = \frac{1}{6^4}$.

$\square$

# 5 Algorithm

Here we briefly discuss how to solve problems of the form:

$$\min_{Z \in \mathbb{R}^{m_1 \times m_2}} \mathcal{L}(Z) + \lambda \Omega_{k,q}(Z). \tag{23}$$

Although convex, this problem can be computationally challenging. We present a working set algorithm to approximately solve such problems in practice.

## 5.1 A working set algorithm

Given a set $\mathcal{S} \subset \mathcal{G}_k \times \mathcal{G}_q$ of pairs of row and column subsets, let us consider the optimization problem:

$$\min_{Z^{(IJ)} \in \mathbb{R}^{m_1 \times m_2}} \mathcal{L}\Big( \sum_{(I,J) \in \mathcal{S}} Z^{(IJ)} \Big) + \lambda \sum_{(I,J) \in \mathcal{S}} \|Z^{(IJ)}\|_* \quad \text{s.t.} \quad \text{Supp}(Z^{(IJ)}) \subset I \times J, \ (I,J) \in \mathcal{S} \qquad (\mathcal{P}_\mathcal{S})$$

Let $(\widehat{Z^{(IJ)}})_{(I,J) \in \mathcal{S}}$ be a solution of this optimization problem. Then, by the characterization of $\Omega_{k,q}(Z)$ in (1), $Z = \sum_{(I,J) \in \mathcal{S}} \widehat{Z^{(IJ)}}$ is the solution of (23) when $\mathcal{S} = \mathcal{G}_k \times \mathcal{G}_q$. Clearly, it is still the solution of (23) if $\mathcal{S}$ is reduced to the set of non-zero matrices $\widehat{Z^{(IJ)}}$ at optimality often called *active* components.

We propose to solve problem (23) using a so-called working set algorithm which solves a sequence of problems of the form $(\mathcal{P}_\mathcal{S})$ for a growing sequence of working sets $\mathcal{S}$, so as to keep a small number of non-zero matrices $Z^{(IJ)}$ throughout. Working set algorithms [2, Chap. 6] are typically useful to speed up algorithm for sparsity inducing regularizer; they have been used notably in the case of the overlapping group Lasso of [7] which is also naturally formulated *via* latent components.

These algorithms rely on the structure of the Karush-Kuhn-Tucker (KKT) conditions for optimality. For problem $(\mathcal{P}_\mathcal{S})$, writing $Z^{(IJ)} = U^{(IJ)} \Sigma^{(IJ)} V^{(IJ)}$ the thin SVD of $Z^{(IJ)}$, the KKT conditions are that, for all $(I,J)$ in $\mathcal{S}$,

$$\text{either} \quad Z^{(IJ)} \neq 0 \quad \text{and} \qquad \Pi_{X,I,J}[\nabla \mathcal{L}(Z)]_{IJ} + \lambda U^{(IJ)} V^{(IJ)^\top} = 0 \tag{24}$$

$$\text{or} \quad Z^{(IJ)} = 0 \quad \text{and} \qquad \|[\nabla \mathcal{L}(Z)]_{IJ}\|_{\text{op}} \leq \lambda. \tag{25}$$

The principle of the working set algorithm is to solve problem $(\mathcal{P}_\mathcal{S})$ for the current set $\mathcal{S}$ so that (24) and (25) are (approximately) satisfied for $(I,J)$ in $\mathcal{S}$, and to check subsequently if there are any components not in $\mathcal{S}$ which violate (25). If not, this guarantees that we have found a solution to problem (23), otherwise the new pair $(I,J)$ corresponding to the most violated constraint is added to $\mathcal{S}$ and problem $(\mathcal{P}_\mathcal{S})$ is initialized with the previous solution and solved again. The resulting algorithm is Algorithm 1 (where the routine SSVDTPI is described in the next section). Problem $(\mathcal{P}_\mathcal{S})$ is solved easily using the approximate block coordinate descent of [14] (see also [2, Chap. 4]), which consists in iterating proximal operators. The modifications to the algorithm to solve problems regularized by the norm $\Omega_{k,\succeq}$ are relatively minor (they amount to replace the trace norms by penalization of the trace of the matrices $Z^{(IJ)}$ and by positive definite cone constraints) and we therefore do not describe them here.

Determining efficiently which pair $(I,J)$ possibly violates condition (25) is by contrast a more difficult problem that we discuss next.

## 5.2 Finding new active components

Checking whether (25) holds amounts to check whether $\arg\max_{(I,J) \in \mathcal{G}_k \times \mathcal{G}_q \setminus \mathcal{S}} \|[\nabla \mathcal{L}(Z)]_{IJ}\|_{\text{op}}$ is smaller than $\lambda$, and if not to find $(I,J)$ that violates this condition. Since the condition is satisfied for $(I,J) \in \mathcal{S}$, this corresponds to solving the following sparse singular value problem

$$\max_{a,b} \quad a^\top \nabla \mathcal{L}(Z) b \quad \text{s.t.} \quad ab^\top \in \mathcal{A}_{k,q}. \qquad (k,q)\text{-linRank-1}$$

This problem is unfortunately NP-hard since rank 1 sparse PCA problem is a particular instance of it (when $\nabla \mathcal{L}(Z)$ is replaced by a covariance matrix), and we therefore cannot hope to solve it exactly

---

**Algorithm 1** Active set algorithm

---

**Require:** $\mathcal{L}$, tolerance $\epsilon > 0$, parameters $\lambda, k, q$

    Set $\mathcal{S} = \varnothing, Z = 0$

    **while** $c = \texttt{true}$ **do**

        Recompute optimal values of $Z, (Z^{(IJ)})_{(I,J) \in \mathcal{S}}$ for $(\mathcal{P}_{\mathcal{S}})$ using warm start

        $(I, J) \leftarrow \texttt{SSVDTPI}(\nabla\mathcal{L}(Z), k, q, \epsilon)$

        **if** $\|[\nabla\mathcal{L}(Z)]_{I,J}\|_{\text{op}} > \lambda$ **then**

            $\mathcal{S} \leftarrow \mathcal{S} \cup \{(I, J)\}$

        **else**

            $c \leftarrow \texttt{false}$

        **end if**

    **end while**

    **return** $Z, \mathcal{S}, (Z^{(IJ)})_{(I,J) \in \mathcal{S}}$

---

with efficient algorithms. Still, sparse PCA has been the object of a significant amount of research, and several relaxations and other heuristics have been proposed to solve it approximately. In our numerical experiments we use a basic truncated power iteration (TPI), also called TPower, GPower or CongradU in the PSD case [8, 16, 10], which has been proved recently by [16] to provide accurate solution in reasonable computational time under RIP type of conditions. Algorithm 2 provides a natural generalization of this algorithm to the non-PSD case. The algorithm follows the steps of a power method, the standard method for computing leading singular vectors of a matrix, with the difference that at each iteration a truncation step is use. We denote the truncation operator by $T_k$. It consists of keeping the $k$ largest components (in absolute value) and setting the others to $0$.

---

**Algorithm 2** SSVDTPI: Bi-truncated power iteration for $(k, q)$-linRank-1

---

**Require:** $A \in \mathbb{R}^{m_1 \times m_2}$, $k, q$ and tolerance $\epsilon > 0$

    Pick a random initial point $b^{(0)} \sim \mathcal{N}(0, I_{m_2})$ and let

    **while** $|a^{(t)\top} A b^{(t)} - a^{(t-1)\top} A b^{(t-1)}| / |a^{(t-1)\top} A b^{(t-1)}| > \epsilon$ **do**

      $a \leftarrow A b^{(t)}$   $\backslash\backslash$ Power

      $a \leftarrow T_k(a)$   $\backslash\backslash$ Truncate

      $b \leftarrow A^\top a$   $\backslash\backslash$ Power

      $b \leftarrow T_q(b)$   $\backslash\backslash$ Truncate

      $a^{(t+1)} \leftarrow a/\|a\|_2$ and $b^{(t+1)} \leftarrow b/\|b\|_2$ $\backslash\backslash$ Normalize

      $t \leftarrow t + 1$

    **end while**

    $I \leftarrow \text{Supp}(a^{(t)})$ and $J \leftarrow \text{Supp}(b^{(t)})$

    **return** $(I, J)$

---

Despite poor computational properties predicted for the worst cases by the theory, in practice Algorithm 2 turns out to perform well, and we observe the linear convergence rate predicted by the theory of [16] very often. We stress that Algorithm 1 is by design robust to certain errors of Algorithm 2. For instance, if an incorrect component is added to $\mathcal{S}$ at some iteration, but the correct components are identified later, the algorithm will eventually shrink the incorrect components to $0$. One of the causes of failure of TPI type of methods is the presence of a large local maximum in the sparse PCA problem corresponding to a suboptimal component; incorporating this component in $\mathcal{S}$ will reduce the size of that local maximum, thereby increasing the odds of selecting a correct component the next time around. If however the heuristic fails at some point to find components that violate (25), we can not guarantee that we have reached global optimality.

## Footnotes

[1]see [13] chapter 14.