[Reviews · NeurIPS 2014]

Submitted by Assigned_Reviewer_25

This paper gives a new convex relaxation for sparse matrix factorization. They show the new relaxation has better denoising performance and lower statistical dimension compared to traditional l_1 or trace norms. However, the new convex relaxation is hard to compute. The paper also involves many potential applications, experiments on denosing performance and application to predict drug-target interactions.

The new (k,q) trace norm is interesting and can potentially be useful. The main concern of the reviewer is the hardness in computing the (k,q) trace norm. The authors present a reasonable algorithm to compute this norm heuristically, but it is not clear whether the theoretical analysis in Sections 3 and 4 still applies. The drug-target interaction experiment looks very interesting, however the reviewer is not familiar enough to know whether the algorithm significantly improves the state-of-art result there.

For the applications in Section 2.3. The (k,q)-trace norm is a reasonable heuristic in these settings, however it would be more convincing if there could be a statement like "if (k,q)-trace norms can be computed (say using some oracle) then we can solve subspace clustering/sparse PCA under certain assumptions".
Summary: This paper gives a new convex relaxation for sparse matrix factorization. The new has good statistical properties, however it is hard to compute.

Submitted by Assigned_Reviewer_26

In this paper, the authors proposed a new approximation of matrix rank and the corresponding norm for solving sparse matrix factorization problems. The performance of this proposed trace norm is analyzed in denoising problem, and its bound of the statistical dimension is also computed. The authors also designed an algorithm to solve the optimization problem with low-rank regularization using this norm. Experimental results are reported to evaluate the effectiveness of the proposed methods.
The proposed work in this paper is interesting. The idea of designing the (k,q)-rank and (k,q)-SVD is original and natural. The analysis of statistical dimension of the proposed trace norm justifies the performance of the algorithm in theory. The experimental results show the effectiveness compared with other norms. The only concern about this paper is the author should report experimental results on real data to show the power of the norm, which is more convincing.
Summary: The proposed idea is fairly novel and interesting.

Submitted by Assigned_Reviewer_29

This paper proposes an atomic norm for estimating low-rank matrix whose factors are sparse. The authors show that the proposed atomic norm can be expressed as additive decomposition of a given (or parameter-) matrix into a sum of low-rank matrices, each of which is supported on at most k rows and q columns. The authors show a "slow rate" for the denoising performance of the proposed norm and compare an upper bound for it with a lower bound for another norm defined as a convex combination of the trace norm and the L1 norm. The method is applied to a drug-target-interaction-prediction problem and shows a AUC score better than the one predicted in the previous paper.

The strength of the paper is that it tackles a problem that have wide applications (bilinear sparse regression, subspace clustering, sparse PCA) with a promising approach. The theory and experiments also seem to show the advantage of the proposed atomic norm over simply using the trace norm, L1 norm, or the convex combination of them.

The weaknesses of the paper are as follows.
- the theories (denoising bound and statistical dimension) are both limited to rank one.
- optimization algorithm could be improved (it uses a generalization of sparse PCA algorithm [22] to compute the dual norm). Could the observation that the proposed norm reduces to the k-support norm for m_2=1 be used to develop a fast algorithm as in [2]?
- Details of the drug-target-interaction experiment are missing. How were the training and test sets seprated? How sensitive was the result to the choice of k and q.
- There were many typos. Sometimes it was hard to understand what the authors mean. For example, "By contrast to sequential approaches that estimate the principal components one by one [12], this formulation requires to find simultaneously a set of complementary factors." (line 164). What do you mean by complementary?

Additional comments:
- line 306: K cannot be PSD because K is in general asymmetric.
- Does the theoretical guarantee in [22] extend to the proposed asymmetric generalization of TPI algorithm?
Summary: An atomic norm formulation for sparse and low-rank matrix factorization. Nice theory and application.
Author Feedback
Author rebuttal: We first would like to thank the reviewer for all their comments.

To reviewer 25:
We agree that the theory we propose is only valid for the global optimum of the problem that we consider. As the reviewer will appreciate, a theoretical analysis of the heuristic we propose might be significantly more challenging. We nevertheless believe that our theoretical results provide a motivation to try and solve the a priori difficult convex problem we aim to solve. It may very well be the case that this problem is in practice not so hard to solve, and yields better estimators than other formulation currently used. This is in fact what our experiments suggest empirically. Note that, in practice the computational bottleneck is the rank-1 sparse PCA, but rank-1 sparse PCA has been the object of quite a bit of research recently, and already the truncated power iteration method works often well [24] and has low complexity, without mentioning other relaxations proposed for it. Our work thus proposes to leverage this existing literature.
Note that there are other settings in machine learning where the optimization problem considered cannot be solved exactly and where heuristics often very clearly suboptimal are used with success: spectral clustering, variational inference, non-convex optimization...

To reviewer 26:
The reviewer says that the algorithm should be applied to real data, but section 6.2 is precisely an application to real data.
Moreover, the reviewer certainly appreciates that the assumption made on the data is more subtle that plain sparsity or low rank, and that in particular the assumption that k and q are fixed might not exactly match a specific data set. In that respect the work that we propose is a bit more theoretical and it is difficult to present both theoretical results and a full blown application in the same paper.

To reviewer 29:
Concerning the weaknesses pointed out by the reviewer:
- Note that the slow rates results of section 3 corollary 1 do not apply only the rank one case, as suggested by the reviewer, but to the general rank r >=1 case. The statistical dimension results indeed only apply to the rank one case; our empirical results suggest that the statistical dimension increases roughly linearly with the (k,q)-rank, although we have no formal proof of it.
- Although the (k,q)-trace norm can be computed efficiently, we believe the fast algorithms proposed in [2] cannot be generalized to the matrix case because this would contradict the fact that sparse PCA is NP-hard. Techniques from [2] could certainly be used in a bi-convex algorithm (where we alternatively optimize the left and right factors) that could be used as a subroutine, but which would not be guaranteed to converge to the global minimum. In fact, one can show that the bi-convex approach leads to algorithms that are a "soft versions" of truncated power iterations. We agree it would be interesting to test this approach in the future and compare it to the truncated power iteration subroutine we used.
- We refer to Yamanishi et al. [23] for the details of the experiments. The drugs, on the one hand, and targets, on the other hand, were randomly split in 5 folds. Training pairs are all pairs in four or the folds, while test pairs are all pairs in the remaining fold. We did not investigate in detail the influence of k and q as this was just a simple attempt to check how the method would behave.
- What we mean by complementary: we say that a set of factors (rank-1 matrices) are more complementary to explain a target matrix if their sum is closer to the target matrix. What we want to express in this sentence is that, given a fixed number of factors that must approximate a target matrix, it is probably better to learn them simultaneously (as we propose) than to greedily find them iteratively, as most existing sparse PCA method do.
- Regarding line 306: We believe the reviewer misread what we wrote: we certainly don't mean to say that K is always psd, but we wanted to say “When k=q and K is PSD, the problem reduces to rank 1 sparse PCA.”
- Concerning whether the guarantee of [22] extends to the asymmetric case: this is an excellent question, which we did not try to prove rigorously. Note however, that if the algorithm choses a support I of size k for u then the result of [22] implies that the best support J(I) is found for v, and, vice versa, given a support J that is fixed for v, the support I(J) that is found is optimal for u, i.e. each of the support is optimal given the other. This provides a guarantee of reaching at least a local optimum. If there is only a single pair (I,J) such that J=J(I) and I=I(J) then this proves that we have the optimal pair.